# Maintenance Performance in the Age of Industry 4.0: A Bibliometric Performance Analysis and a Systematic Literature Review

**DOI:** 10.3390/s23031409

**Published:** 2023-01-27

**Authors:** Sylwia Werbińska-Wojciechowska, Klaudia Winiarska

**Affiliations:** Faculty of Mechanical Engineering, Wroclaw University of Science and Technology, Wyspianskiego 27, 50-370 Wroclaw, Poland

**Keywords:** maintenance, Maintenance 4.0, Industry 4.0, data-driven decision making, Operator 4.0, virtual reality, augmented reality, cyber–physical system, cybersecurity, systematic review

## Abstract

**Featured Application:**

**This article is focused on a literature review to provide a valuable resource for understanding the latest developments in the Maintenance 4.0 approach. The conducted research will be helpful for many people, including maintenance managers, maintenance engineers, and researchers, who are interested in the issues of maintenance performance in the context of Industry 4.0 technologies implementation. The conducted literature review intends to introduce the readers to the major up-to-date theory and practice in Maintenance 4.0 main research directions. The presented study makes it possible to identify the thematic structure related to maintenance performance. In addition, it shows which topics from the studied scientific area are the most investigated in a given country/region. At the same time, the conducted analysis allowed the development of future research directions in the areas identified as research and knowledge gaps.**

**Abstract:**

Recently, there has been a growing interest in issues related to maintenance performance management, which is confirmed by a significant number of publications and reports devoted to these problems. However, theoretical and application studies indicate a lack of research on the systematic literature reviews and surveys of studies that would focus on the evolution of Industry 4.0 technologies used in the maintenance area in a cross-sectional manner. Therefore, the paper reviews the existing literature to present an up-to-date and content-relevant analysis in this field. The proposed methodology includes bibliometric performance analysis and a review of the systematic literature. First, the general bibliometric analysis was conducted based on the literature in Scopus and Web of Science databases. Later, the systematic search was performed using the Primo multi-search tool following Preferred Reporting Items for Systematic Reviews and Meta-Analyses (PRISMA) guidelines. The main inclusion criteria included the publication dates (studies published from 2012–2022), studies published in English, and studies found in the selected databases. In addition, the authors focused on research work within the scope of the Maintenance 4.0 study. Therefore, papers within the following research fields were selected: (a) augmented reality, (b) virtual reality, (c) system architecture, (d) data-driven decision, (e) Operator 4.0, and (f) cybersecurity. This resulted in the selection of the 214 most relevant papers in the investigated area. Finally, the selected articles in this review were categorized into five groups: (1) Data-driven decision-making in Maintenance 4.0, (2) Operator 4.0, (3) Virtual and Augmented reality in maintenance, (4) Maintenance system architecture, and (5) Cybersecurity in maintenance. The obtained results have led the authors to specify the main research problems and trends related to the analyzed area and to identify the main research gaps for future investigation from academic and engineering perspectives.

## 1. Introduction

One of the most important issues in ensuring the high availability and reliability of technical facilities and systems is maintaining them in operational conditions [1]. Maintenance has recently been an important area of interest and research for engineers and managers, as improperly maintained equipment can lead to more frequent failures of facilities and their components, low operational efficiency, or delays in meeting operational schedules [2,3]. A poorly chosen or planned maintenance strategy for any equipment can result in, among other things, obtaining products of questionable quality, decreasing energy efficiency in some areas, or under/over utilization of maintenance personnel [4,5]. Following this, more and more companies are taking steps to improve the efficiency of the maintenance function of their physical assets [1,6,7]. In addition, the problem of maintenance cost modeling and optimization is gaining attention [8]. 

Recently, there has been a lot of research and publications in the field of maintenance models and decision-making techniques aimed at improving the efficiency of the maintenance process (for an overview, see, for example, [1,9]). Known solutions have evolved from Maintenance 1.0 to Maintenance 4.0 [9]. On the other hand, organizations strive to improve their maturity in implementing maintenance strategies. According to the authors of a report [10] that surveyed the implementation of maintenance strategies in companies in Belgium, Germany, and the Netherlands, only 11% of respondents (a total of 280 people) indicated that their companies had reached Level 4.0. Following the report [11], the global predictive maintenance market size was valued at USD 3.18 billion in 2018. In addition, according to the consulting group Next Move Strategy Consulting [12], the global predictive maintenance market is expected to register a CAGR (Compound Annual Growth Rate) of 30.47% between 2020 and 2030. Therefore, it is imperative to examine the main trends occurring in the maintenance area in the context of Maintenance 4.0. 

A preliminary analysis of the resources from such databases as Web of Science and Scopus allows us to state that more than 50 review papers on predictive maintenance (PdM)/Maintenance 4.0 have been published in the last decade (Note: The main search procedure was performed for the key term: “Maintenance 4.0 review”, and the results were limited to the relevant time period and research field). On the one hand, the growing number of publications focused on reviewing recent developments for Maintenance 4.0 confirms the relevance of the issue and the potential for its development. On the other hand, it shows how much the subject has developed in one decade in many aspects of industry sectors. A short summary of recent papers focused on the Maintenance 4.0 literature reviews is presented in Table 1. 

This briefly presented background of conducted and published literature reviews on predictive maintenance/Maintenance 4.0 allowed us to conclude that there is currently a lack of studies that summarize recent developments in a cross-sectional manner. Most of the conducted reviews focus on a particular application area or the industry sector (see, e.g., [13,14,15]). The literature specializes in specific topics, treating them separately (e.g., [16,17]). As a result, treating this as a research gap, the authors decided to provide a complete review of the existing literature to present an up-to-date and content-relevant analysis in this field, focusing on both bibliometric performance analysis and a systematic literature review. 

Following this, the research questions are as follows:

RQ1: What are the current trends in Maintenance 4.0 approaches, and how have these trends evolved over the last decade?

RQ2: What are the future research directions and perspectives in Maintenance 4.0 in the context of the defined application fields?

Therefore, the article aims to develop a literature review in the area of Maintenance 4.0 main application fields, including (1) bibliometric performance analysis of research works from the period 2012–2022 being published in two scientific databases—Web of Sciences and Scopus, and (2) systematic analysis using the Preferred Reporting Items for Systematic Reviews and Meta-Analyses (PRISMA) method, aimed at summarizing and identification of the main research areas in the identified application fields. Following this, the main contributions of this paper include the following:A summary of the research developed in Maintenance 4.0 application fields in the last decade, focusing on (a) augmented and virtual reality, (b) system architecture, (c) cybersecurity, (d) data-driven decision, and (e) Operator 4.0;Identification of research gaps and knowledge gaps in the identified application fields of the Maintenance 4.0 approach.

In conclusion, the article is organized into seven sections (Figure 1). After the Introduction (Section 1), the Theoretical Background (Section 2) introduces the concept of Industry 4.0 and discusses the evolution of maintenance approaches, focusing on the Maintenance 4.0 concept. Review methodology (Section 3) explains the main methods used for the review. This section also describes the strategy used for the literature search process performance and criteria that were applied to assess the relevance of analyzed documents. Section 4 describes the main results of conducted bibliometric performance analysis in the macro-view and for the selected papers on the five identified application fields. Later, Section 5 focuses on presenting the results of the identified application fields. Section 6 provides a discussion of the obtained results. Here, the literature research and knowledge gaps are also identified. The last part contains conclusions (Section 7) with a summary of contributions, limitations definition, and recommendations for future studies’ presentation. 

## 2. Theoretical Background

### 2.1. Industry 4.0

The first studies on the Industry 4.0 emerged in 2011 [43]. They highlighted the new high-tech techniques, such as Internet of Things (IoT) platforms, advanced human–machine interfaces, smart sensors, big data-based analytics, augmented reality-based solutions, and the concept of the smart factory have become part of industrial production [37]. In line with the observed rapid technical and technological development of the global economy, the so-called fourth industrial revolution started only 42 years after the third revolution, relatively short compared to the 99 years recorded between the second and third industrial revolutions (Figure 2) [44]. At the same time, it posed new challenges to managers in terms of building the so-called Operator 4.0 competence or effectively implementing modern technologies in practice. 

The Fourth Industrial Revolution is the next stage of socio-economic development. This revolution is associated with [44,45]:Widespread digitalization and the provision of constant communication between people, people with devices, and devices between one another;An increase in the implementation of disruptive innovations;A leap in the efficiency of the socio-economic system performance;The development of machines capable of autonomous operation through the use of artificial intelligence (AI);The implementation of modern communication using technology and the capabilities of modern networks throughout the supply chain;Adaptive automation;The use of intelligent approaches to information processing;The use of future-oriented techniques.

The term Industry 4.0 refers to the combination of several significant innovations in digital technology. These technologies include, among others, advanced robotics and artificial intelligence, digital manufacturing (including 3D printing), software as a service, and other new business models (robot as a service, machine as a service, software as a service, etc.) as well as support in decision-making processes [46]. In addition, in the recent literature, the term smart industry is used concerning the Industry 4.0 concept. Both terms are often used interchangeably (see, e.g., [47]). However, the first works focused on the smart industry concept mostly referred to industrial IoT technologies use. The main goal was to interconnect various industrial objects through sensors, GPS devices, radio frequency identifiers, actuators, and other wireless and mobile devices [47]. Recent works extend this approach by implementing other Industry 4.0 technologies in the context of smart industry performance, such as big data, digital twins, or artificial intelligence (see, e.g., [48]). Indeed, smart industry enables organizations to maximize the yield from existing operational capabilities and to develop the next generation of operational capabilities necessary to compete in a digital economy [49]. As a result, the term Industry 4.0 may be perceived in the context of developing smart connections between products, machines, and people using the latest technologies to provide more efficient, intelligent, and aware factory performance. It is also connected with the e-commerce sector’s rapid development [50,51] (Figure 3). 

It has become commonplace to use smartphones or other mobile devices and platforms that use algorithms to drive motor vehicles (including navigation tools, ride-sharing apps, delivery and transportation services, and autonomous vehicles) in daily operations and to embed all these elements in an interoperable global value chain shared by many companies from many countries (Figure 4) [44,52].

**Figure 3 sensors-23-01409-f003:**
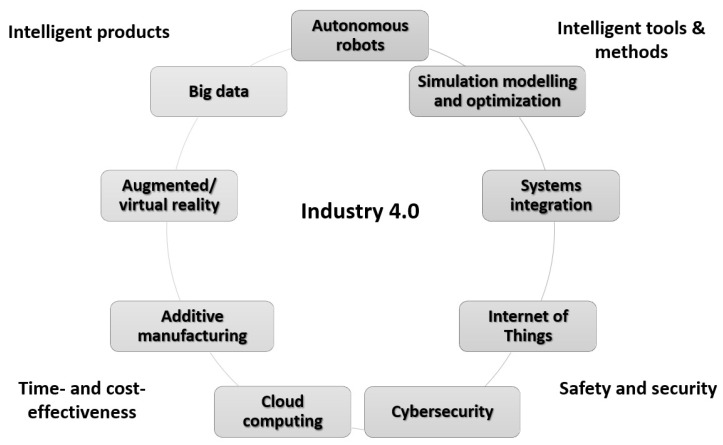
Technologies prevalent in Industry 4.0. Source: own contribution based on [53].

Simultaneously, the high complexity, automation, and flexibility of the so-called smart factory bring new challenges regarding reliability and safety [54]. The digital factory requires the exchange of data between machines or the exchange of data on production and operation and is intended to optimize costs, improve availability and reliability, or ensure an adequate level of overall equipment effectiveness (OEE) [55]. Therefore, to gain maximum benefit from the implementation of the Industry 4.0 concept, the production system and the factory itself must be connected internally (using the Intranet) throughout the organization and externally with suppliers and customers, so that important information and data can be exchanged (promptly) within the complete supply chain (using the Internet) [56,57]. Further, networks can be developed into connected factories operating in different regions. In this approach, business data are combined, compared, and processed in one place. At the same time, such cooperation can concern the level of a single department or the entire business organization [52]. The highest level of organizational maturity is the so-called Industry 4.0 enterprise, in which we combine the supply chain with product design and after-sales service using Industry 4.0 technologies. This way, product customization can be achieved in a highly flexible production environment [54].

Smart production is designed based on a modular structure, where production facilities and logistics systems are organized without human intervention [58]. Production is supervised using a number of intelligent sensors, cameras, and indicators. Where information is not exchanged online, data are stored in the device’s memory and exchanged on demand. In addition to data on the condition of the product itself, data on the condition of the machinery and equipment operating in the factory are also collected [59,60]. The collection and analysis of this data lead to the Maintenance 4.0 concept, which is described in detail in Section 2.3 [61]. 

### 2.2. Maintenance Approaches Evolution

Currently, in the available literature, we can distinguish many definitions of the concept of technical maintenance [1]. For the purpose of this paper, the literature survey carried out is based on the definition proposed in the European standard BS EN 13306: 2017 [62], where “*maintenance/operation is a set of all technical, organizational and managerial activities during the life cycle of an object, the purpose of which is to maintain or renew the state in which it can be used to fulfil the required function*”. A similar definition is given in IEC 60300-3-10: 2001 [63], in which maintenance of a technical system is defined as “*all activities necessary to preserve/maintain or restore a specific state of an object*”. Maintenance management, on the other hand, according to BS EN 13306: 2010 [62], is defined as “*all management activities that determine the objectives and strategies of maintenance and their implementation through appropriate tools, such as maintenance planning, control and supervision, improvement of methods in the organization, together with consideration of the economics of operation*”.

Based on the definitions presented, one may state that the main objectives of the maintenance of technical systems are related to [64]:Ensuring the basic functional parameters of the facility (e.g., availability, efficiency, reliability);Efficient management of resources to provide the required durability of equipment;Ensuring efficient use of resources, energy, and raw materials/replaceable parts;Ensuring the safety of a technical facility, people, and the environment;Taking into account the financial requirements of the implemented operation process.

The achievement of such objectives is based on the use of appropriate maintenance approaches. The evolution of known approaches to maintenance that have been developed over the last fifty years is shown in the scheme included in Figure 5. At the same time, a primary classification of known and widely used maintenance strategies is given in EN 13306 [62]. A recent review of the literature on technical maintenance of systems and facilities can be found, for example, in works [35,65,66], and an analysis of the development of maintenance philosophies can be found in [67]. A general classification of maintenance strategies can be found in [1]. 

The first approach to maintenance (Maintenance 1.0), often referred to as the ‘run to failure’ or Corrective Maintenance (CM) strategy, was prevalent between 1940 and 1960 [1]. CM is a reactive strategy and refers to all unplanned maintenance activities performed to restore the system’s operability after a failure by carrying out repair operations or replacing failed components, implying that there is no optimization in the maintenance activities undertaken concerning economic or reliability criteria. On the other hand, this maintenance strategy is still popular due to the low cost of its implementation [1].

When it is necessary to avoid a system failure during operation, especially when such an event is costly and/or dangerous, it is important to carry out planned maintenance activities [1,8]. Therefore, the Maintenance 2.0 approach related to Preventive Maintenance (PM) is being implemented. PM, according to MIL-STD-721C [68], refers to “*all activities undertaken to maintain a facility/system in a specified condition by systematically inspecting, detecting and preventing incipient failures, with the objective of reducing the probability of failure or slowing the degradation processes of a system in operation*”. In general, the approach is geared toward optimizing the length of maintenance intervals in relation to the wear and tear process of the investigated system. In this area, time-based PM and on-condition maintenance (CBM) strategies are the most frequently mentioned in the literature [9]. Furthermore, the differences between CM and PM are presented in [69], and a comparison of the leading maintenance strategies is in [70]. 

A periodic maintenance strategy involves planned and periodic repairs/replacements of equipment. It is still one of the predominant maintenance strategies used in practice for those technical assets for which it is impossible to implement diagnostic measures, e.g., for technical or economic reasons [71]. For example, more information can be found in the works [72,73]. 

A condition-based maintenance approach is considered the first maintenance strategy to be included in Maintenance 3.0 [9]. CBM is based on monitoring the parameters that define the technical condition of a system or its components using diagnostic methods/measures [74]. For selected cases, the CBM strategy even offers the possibility to implement maintenance activities just before the failure of the system/its components occurs. Thus, CBM can be regarded as a method used to reduce the uncertainty of maintenance activities for technical systems [75]. A literature review of CBM policies is presented, for example, in the works [74,76,77].

Another maintenance policy, which in the literature is often considered synonymous with the CBM concept, is predictive maintenance (PdM) [17]. This maintenance policy is used in those sectors where reliability is of strategic importance, such as nuclear power, transport, or energy industry solutions [1,9]. Its main task is to predict faults or failures in a degrading system to optimize maintenance tasks by monitoring system operating conditions to detect any signs of wear and tear that lead to component failure [10]. The PdM program aims to track component degradation/wear using a methodology that ensures the detection of any impending failure [78]. Some commonly used monitoring and diagnostic techniques include vibration monitoring, thermography, or visual inspection [79].

A proactive approach to maintenance in line with the Maintenance 4.0 concept is currently the most technologically advanced form of maintenance for technical systems. In the literature and practice of the issue under study, it is often emphasized that the Maintenance 4.0 concept is a practical implementation of PdM and SMART PdM solutions [80,81]. Therefore, references can be found to the so-called Predictive Maintenance 4.0 concept.

### 2.3. Maintenance 4.0 and the Leading Technologies

If the main elements of Industry 4.0 are the Industrial Internet of Things (IoT), cloud computing, and technologies such as augmented reality (AR) and virtual reality (VR), Maintenance 4.0 is based on the implementation of these technologies in the company’s maintenance practices [82]. Maintenance 4.0 is thus about *using smart technologies to improve daily factory operations* [83]. The aim is to maximize uptime by eliminating unplanned, reactive maintenance activities. One of the concepts used here is the Internet of Things, which takes machine-to-machine technology to the next level by including a third element: data. According to [84], all machine data are to be available on a single virtual network, giving manufacturers the ability to aggregate and analyze data to generate better predictive analytical models. For more information, the author recommends reading, for example, [10,17]. On the other hand, a literature review on Maintenance 4.0 and the smart industry can be found among others in [15,36,42].

Against this background, it is possible to define the directions for developing the Maintenance 4.0 concept in business practice as a general overview observed in polish companies (mainly from the automotive sector) (Figure 6). In the first step of implementing the solutions of the Maintenance 4.0 concept, investments are made in the so-called intelligent components—a system of diagnostic sensors, which usually refers to the possibility of vibration analysis performance [32,52]. In the next step, a monitoring and diagnostic system are developed for the selected machine, and PdM-type systems based on fundamental, predictive analyses are introduced The smart factory is already at the level of developing dynamic self-diagnostic systems, the so-called SMART PdM—based on acquired diagnostic data, failure prediction, and automation of maintenance processes. The level of connected factories includes designing and implementing a so-called centralized control system for maintenance processes based on solutions such as asset sharing or IoS business models [85]. 

The highest level of organizational maturity allows the implementation of intelligent maintenance methodologies already at the product design level [85].

Based on the presented developments in the implementation of Maintenance 4.0, it can be concluded that this approach allows the design and implementation of so-called self-aware (self-diagnose) and self-service (self-service) equipment systems that can self-assess their technical condition and degradation process and use information from other machines to make intelligent maintenance decisions [60]. The Deloitte report classifies the technologies that underpin PdM 4.0 under five categories [53]. These are sensors, networking, integration, augmented intelligence, and augmented behavior (Figure 7). 

A key issue, therefore, is the continuous monitoring and analysis of the physical asset network, which enables [88]:Predicting and notifying of potential failures/damages;Maintenance scheduling and planning of spare parts requirements;Automation of specific maintenance tasks.

In summary, Maintenance 4.0 encompasses a holistic view of data sources, how they are combined, collected, analyzed, and are recommended actions to provide digital support to the function (reliability) and value (management) of assets. As a result, a holistic approach enables effective plant-wide communication between machine operators, maintenance and engineering teams, and management, allowing informed decisions and better utilization of resources [89]. In addition, implementing a holistic approach to predictive maintenance provides that individual components are assessed for their value in the entire production chain and sensors are applied accordingly. Indeed, a wide range of complex, interconnected assets must be considered for interdependencies rather than their singular function alone [90]. Properly developed, a holistic approach is to be shown to ensure the maximum potential for early warning analysis and root cause identification in technical systems [91].

As a result, it is essential to analyze the main trends in developing the Maintenance 4.0 concept in practice and literature. This issue is the subject of the next sections of this paper.

## 3. Review Methodology 

The main goal of the conducted review is to investigate the five main research areas developed under the Maintenance 4.0 concept. As part of our literature review, we analyzed, evaluated, and discussed scientific publications from prestigious databases related to engineering fields. Therefore, a two-phase review methodology was implemented, incorporating the following:**Bibliometric analysis of the literature within the scope of application.**

The main focus of the bibliometric analysis of the literature is to present a macro-view of Maintenance 4.0 and its leading research areas investigated over time. The first phase also allowed for properly defining the main inclusion criteria for SLR performance in the second step of the reviewing methodology.


**Systematic analysis of the selected papers within the scope of application.**


The main focus of systematic analysis is to define the main aspects and trends occurring in the area of Maintenance 4.0 research. 

The first of these methods is a form of quantitative analysis, while the second is both quantitative and qualitative. As a whole, this methodology adequately presents the full spectrum of publications connected to the Maintenance 4.0 concept, both in quantitative and qualitative terms. In addition, it allows for presenting a macro and micro view of the investigated issues. Following this, the research procedure and review strategy phases are shown in Figure 8.

First, we defined research questions that helped with keyword selection. In the next step, the main keyword, “Maintenance 4.0”, was described to identify in the broadest possible way the relevant articles related to the maintenance issue in the context of the Industry 4.0 concept. The next steps are connected with the performance of meta-analysis and systematic analysis of the identified literature. A detailed description of the conducted research procedure is presented in the following subsections.

### 3.1. Meta-Analysis of the Literature

A meta-analysis is *the statistical pooling of data across studies to generate summary estimates of effects* [92]. This paper conducted such an analysis based on a bibliometric performance analysis approach following the PRISMA guidelines, as defined in [93,94]. PRISMA is an evidence-based approach for reporting in systematic reviews and meta-analyses [92]. Therefore, it was used in both phases of the conducted literature review in Maintenance 4.0. 

Bibliometrics is a branch of scientometrics that uses mathematical and statistical methods to assess scientific activities’ performance. The bibliometric analysis allows us to study the networks formed around the most representative keywords and presents how citations, scholars, affiliations, counties, and publications indicate the importance of specific topics in the field of research. At the same time, we can see a noticeable increase in interest in bibliometric studies in science (see, e.g., [95,96,97,98]). As a result, the analysis’s objective is to identify the main trends in Maintenance 4.0 research through bibliometric analysis. 

A preliminary bibliometric analysis was conducted using two available databases: Web of Science [99] and Scopus [100]. The investigation was completed in October 2022, and the search term was the keyword “Maintenance 4.0”. 

The selection of two databases, Web of Science and Scopus, is related to the fact that both of these databases have similar bibliographic attributes, such as literature searching and citation analysis of bibliometric records. As a result, it was considered that an analysis of both scientific databases would provide a compatible and complementary view of Maintenance 4.0 issues.

At first, the Web of Science database was under investigation. The initial search procedure was based on the following search term “Maintenance 4.0” regardless of where it occurred (filtering by “all fields”). The first step of the searching procedure provided the opportunity to identify 4944 relevant records. In the next step, an objective screening was performed based on the title and keywords. To evaluate the eligibility, the research team analyzed the title and keywords of publications. For this purpose, we have defined criteria for exclusion. First, to focus on relatively new problems and technologies, the search results were limited to papers published within the last ten years.

Additionally, articles written in English were considered. In addition, the search was limited to citation topics related to production, logistics, or maintenance (non-topically related publications were excluded). Indeed, papers that do not have a production/logistics/factory focus were excluded from the further analysis (for example, we eliminated documents related to medicine, biology, and agriculture). The indicated exclusion criteria were included as subject area filters within the WoS search string. Indeed, within the framework of the selection process, the authors have selected the subject area filters available in the WoS database that are related to these three indicated citation topic areas. As a result, such citation topics as nutrition and dietetics, urology and nephrology, dairy and animal sciences, anesthesiology, and dentistry and oral medicine have been excluded from further research. 

Based on these exclusion criteria, 1113 records were identified and subjected to fundamental bibliometric analysis.

The Web of Science database final search engine:
(ALL = (Maintenance 4.0)) AND LA = (English) and Design & Manufacturing or Safety & Maintenance or Friction & Vibration or Telecommunications or Human Computer Interaction or Management or Supply Chain & Logistics or Artificial Intelligence & Machine Learning or Software Engineering or Transportation or Robotics

An analogous bibliometric analysis was carried out for publications included in the Scopus database. The first stage of the study identified 24,603 relevant records (searching by Maintenance 4.0 keyword and filtering by “all fields”). Subsequent filtering by “abstract/title/keywords” reduced the number of records searched to 3747 publications. In the final step, the analysis was again narrowed down to publications from 2013-2023, and only texts in English were included. Non-topically related publications were also excluded. Indeed, the filtered subject areas also have been related to the production/logistic/maintenance issues. Therefore, subjects such as medicine, mathematics, energy, biochemistry, genetic and molecular biology, chemistry, or nursing were excluded from further analysis. 

This allowed 1933 publications to be highlighted, which were then analyzed.

The Scopus database final search engine:
(TITLE-ABS-KEY (Maintenance 4.0) AND LANGUAGE (English)) AND PUBYEAR > 2012 AND (LIMIT-TO (SUBJAREA, “ENGI”) OR LIMIT-TO (SUBJAREA, “COMP”) OR LIMIT-TO (SUBJAREA, “MATH”) OR LIMIT-TO (SUBJAREA, “DECI”) OR LIMIT-TO (SUBJAREA, “MATE”) OR LIMIT-TO (SUBJAREA, “BUSI”) OR LIMIT-TO (SUBJAREA, “SOCI”) OR LIMIT-TO (SUBJAREA, “ECON”) OR LIMIT-TO (SUBJAREA, “PSYC”)) 

As a result of the selection process for both databases, we finally received 1133 records for the Web of Sciences database and 1933 records for the Scopus database. In the next step, the obtained results for both databases were separately subjected to analysis and synthesis. First, the analysis was performed using Mendeley reference manager and Microsoft Excel software. This allows us to perform content-based analysis for trends or frequency of occurrence. Next, we used the clustering method during the analysis based on the use of VOSviewer software [101]. Based on [102], the VOSviewer is a program developed for constructing and viewing bibliometric maps that can be examined in full detail. Following its functionality, we performed a co-occurrence analysis of keywords. The distance-based bibliometric maps reflect the strength of the relation between the selected keywords. Indeed, the cluster analysis results clearly capture the knowledge structure of the research fields. They were also used for adequately selecting inclusion criteria for systematic analysis performance.

### 3.2. Systematic Analysis of the Literature

The review’s second phase focused on systematic analysis (Figure 8). A systematic review is “*connected with identifying, evaluating and interpreting all available research relevant to a particular research question, or topic area, or phenomenon of interest”* [103]. It follows a standard procedure for developing, conducting, and reporting processes, reported in detail in [92,104]. The basis for reporting systematic review conducted by the research team was PRISMA guidelines. The chosen method gives the possibility to properly search and select the relevant scientific literature on the given topic by defining research objectives and providing clear quantification of scientific developments in a specific field of knowledge (see, e.g., [13,105,106]). 

Following this, the next subsections explain the document search and selection process with a definition of eligibility criteria and identification of relevant papers for further investigation.

#### 3.2.1. Collection of Publications for Review 

The literature searching process was based on using the multi-search tool Primo [107], which allowed the analysis of many information resources, including, among others, the ScienceDirect database, Elsevier, Wiley, and Springer publishers’ databases. The literature search was conducted between 2 October 2022 and 10 October 2022. 

The initial search procedure was based on the following “Maintenance 4.0” search term. The first step of the search procedure allowed the identification of 309,994 relevant records. In the next step, to focus on relatively new applications, problems, and technologies, the searches were limited to studies published during the last ten years. Additionally, only documents written in English were considered. Based on these exclusion criteria, 219,725 records were identified and further analyzed based on the filtering and extraction process.

#### 3.2.2. Filtering and Extraction 

The authors focused on filtering studies, considering six inclusion criteria. The criteria were defined based on the results obtained from the first phase of the conducted literature review.

Based on the meta-analysis, we could identify the primary clusters with relationships occurring between critical keywords. With the use of VOSViewer software, the authors conducted an in-depth analysis of the main keywords’ co-occurrence. First, they focus on the 223 most frequently used keywords for the WoS database (keywords occurred at least five times). This analysis made it possible to identify nine main clusters. Within the defined clusters, the first cluster was connected with data collection, analysis, and decision-making processes (58 items selected). The second cluster focused on digitalization and smart maintenance as synonyms of the Maintenance 4.0 term (35 items). The third cluster includes keywords connected with cyber–physical systems and human–machine interactions (31 items). The next cluster focused on maintenance planning and scheduling (26 items). Cluster no. 5 (22 items) and Cluster no. 6 (19 items) regarded virtual and augmented reality. 

Next, a similar analysis for the Scopus database was performed. On the one hand, it confirmed the main research findings obtained from WoS database analysis (for 792 keywords and 12 clusters). Conversely, it allowed identifying another significant cluster connected with cybersecurity (Cluster no. 7, 48 items). The six main inclusion criteria were defined based on this initial scientometric analysis and following the reviews on this topic (e.g., [25,27]). As in the case of the definition of the basic keywords “Maintenance 4.0”, inclusion criteria were defined at a general level. This made it possible to encompass a broad spectrum of investigated problems related to maintenance. The main inclusion criteria were as follows:“data-driven maintenance”;“system architecture”;“Operator 4.0”;“virtual reality”;“augmented reality”;“cybersecurity”;

The screening process had the purpose of filtering out papers that were not related to the main topic. Therefore, the identified records were scanned by title. The filtering process was performed manually by the research team based on the filtering options available in the Primo search tool. The filtering process was carried out separately for each inclusion criteria according to the filtering procedure: All fields = “Maintenance 4.0” AND Title = “inclusion criterion”.(1)

The results of this procedure are presented in Table 2. 

As a result of the filtering process considering the inclusion criteria, 218,918 were eliminated out of the initial 219,725 records (based on Table 2 results). 

In the next step of the filtering and extraction procedure, the search was limited to the following documents: articles (Note: In the Primo search tool, the term “articles” relates to scientific articles published in journals and conference articles published in high-quality proceedings from, e.g., Elsevier’s journals (e.g., Procedia CIRP)), books, and book chapters for a higher data quality.

The last exclusion criterion regarded the type of online databases used. The search procedure was limited to online databases like ProQuest Central, EBSCO, IEEE Electronic Library, Springer (All available), or Elsevier with ScienceDirect. The excluded databases from the further analysis were:Health and Medical collection;Earth, Atmospheric, and Aquatic Science Database;GFMER Free Medical Journals;Environmental Science Database.

This choice was connected with eliminating work unrelated to the production/logistics/factory maintenance areas.

After applying these rejection criteria, the documents were reduced to 517. Moreover, 31 publications were deleted as duplicates. As a result, 486 papers were defined, which were later fully read to identify the most relevant papers during the selection process.

#### 3.2.3. Selection Process 

The authors later examined 486 papers to verify their eligibility for further qualitative and quantitative analysis. The main criterion applied in the full-text research was its relevance to the investigated thematic area and defined groups. The authors evaluated the publications first individually. Later, at research team meetings, we compared team members’ opinions. In case of discrepancies in assessing the paper’s suitability, the team members focused on reviewing the whole document more thoroughly. In addition, the studies that describe maintenance issues concerning, e.g., chemical engineering or medicine applications were excluded. After a consensus between the authors of this systematic review, 272 papers were rejected as being out of scope after reviewing the full document.

Consequently, a total of 214 manuscripts were included for further qualitative and quantitative analysis. Figure 9 represents the flow diagram of the selection of studies according to PRISMA statements. The PRISMA checklist is available in Appendix A.

#### 3.2.4. Content Analysis and Synthesis

The selected papers were further subject to analysis and synthesis. As during the meta-analysis, to carry out this analyses we used Mendeley reference manager and Microsoft Excel software. In addition, VOSviewer software was used for cluster analysis.

First, the bibliometric analysis for the five selected research areas in Maintenance 4.0 was performed. The main results are presented in relation to, among others, authors’ location, publication time, or most frequently used keywords. The results are shown in Section 4.2.

The next step was focused on the main research fields overview. Later, the obtained research outputs were discussed concerning the defined research questions RQ1 and RQ2. The results are presented in Section 5 and Section 6.

## 4. Bibliometric Performance Analysis of the Literature within the Scope of Application

### 4.1. General Bibliometric Performance Analysis

First, the meta-analysis results for the Maintenance 4.0 concept’s macro-view and main research fields were investigated. The data are sourced from the Web of Sciences [99] and Scopus [100], one of the largest scientific literature databases. They were accessed on 2 October 2022. The conducted analysis provided a curated dataset of relevant publications for further state-of-the-art research. 

The publications year-on-year for the analyzed period of time for both datasets are presented in Figure 10. The results confirm a considerable rise in year-on-year Maintenance 4.0 publications, especially in the last five years. The exception is 2020, which annotated a decline in publications in the scientific area under study. This may be due to the pandemic period, in which engineering issues related to the development of technologies such as augmented or virtual reality may have been given lower priority over, for example, research areas such as risk analysis or resilience engineering. 2022 is not finished yet, so we do not have a complete picture of the development trend of the number of publications in a given area. Additionally, 2023 is also excluded from the visualization. 

In the Web of Science database, each paper can be classified according to the publication titles. In the case of the investigated research area, the most frequently selected publication titles, where the documents on Maintenance 4.0 appeared, are presented in Figure 11. During the analyzed time period, the most significant number of publications appeared in the journal *IFAC-PapersOnLine* series (72 articles). Numerous publications can also be found in *Proceedia Computer Science* (50 articles), *Procedia CIRP* (46 articles), and *Applied Sciences* (44 articles). This paper distribution indicates that during the analyzed period of time, researchers were very keen to publish the results of their research at conferences across all topics of, among others, computer science, mechanical engineering, and manufacturing engineering areas. 

The presented analysis can be supplemented with the results obtained from the Scopus database. In the Scopus database, each paper is classified by subject area. In the investigated case, one can distinguish eight subject areas that include the most significant number of publications (Figure 12). The obtained results are compatible with Web of Science database analysis conclusions. The engineering and computer science publications account for 70% of all results analyzed. 

Another important aspect was investigating the main keywords in the analyzed publications. First, the Web of Science dataset was under investigation. The analysis was performed for the keywords which occurred in the database at least ten times. This restriction identified the 97 most popular keywords, which formed five main clusters, with a link strength of 2503 and a total link strength equal to 8762. The results are shown in Figure 13. 

Two clusters for the item ‘predictive maintenance’ and for ‘Industry 4.0’ are particularly noteworthy in the area under review. The results of the strength of the links in the respective clusters are shown in Figure 14 and Figure 15. The strength of linkages indicates a strong interest in Maintenance 4.0 issues in the context of predictive strategy and the Industry 4.0 concept. Previous review publications in the field of Maintenance 4.0 also confirm such results. On the other hand, such keywords, like augmented reality, machine learning, or digital twin, also indicate the high strength of the links and suggest a potential direction for research in the area under consideration. An analysis of the database confirms this conclusion in the context of the publication date. According to the results obtained, the most recent studies are about digital twins, deep learning, or smart factories.

A similar analysis was performed for the Scopus database. The selected 1933 papers were reviewed for highlighting all keywords that have occurred a minimum of 10 times in the database. The obtained results are presented in Figure 16. The clusters for ‘predictive maintenance’ and for ‘Industry 4.0’ are shown in Figure 17 and Figure 18. The obtained results confirm the previously defined conclusions. Additionally, an interesting aspect is the occurrence of a cluster related to keywords such as ‘automated maintenance’, ‘intelligent operations’ or ‘digital storage’. This is confirmed by the emergence of a number of publications that target solutions related to the digitalization and automation of industrial processes.

### 4.2. Bibliometric Performance Analysis of the Papers Selected in the Investigated Five Research Fields

The second step of the conducted bibliometric analysis includes the detailed investigations carried out for selected articles from five thematic groups for Maintenance 4.0 from the last decade.

Two hundred fourteen articles from five analyzed areas were adopted for detailed analysis. Most publications were found for the keywords ‘*augmented reality*’ and ‘*virtual reality’*, which accounted for almost 45% of all the analyzed texts. The number of publications for each of the analyzed search terms was:In the area of *augmented and virtual reality*: 96 publications;In the area of *system architecture*: 41 publications;In the area of *cybersecurity*: 37 publications;In the area of *data-driven decisions:* 23 publications;In the area of *Operator 4.0:* 17 publications.

The analysis of the authors’ and scientific centers’ origins allows us to state that most of the publications from the studied area come from the USA (24 papers), China, and Italy (17 papers per country), Germany (15 papers), and three countries: Poland, Greece, and the United Kingdom (UK) (10 papers per country). The regions of origin of the analyzed publications’ authors are shown in Figure 19.

The analyzed publications were limited in the second step of the adopted methodology to those published during the last decade. The adopted limitation seems to be correct, as the verification of the years in which subsequent articles were published indicates a generally high increase in publication frequency from 2017. As shown in Figure 20, the annual number of publications has been about 30 or higher for the last five years, while in previous years, it did not exceed four articles per year. This suggests that the research issues connected with Maintenance 4.0 are far from being exhausted, and its popularity among researchers should be still rising. It is safe to say that further developments regarding this field of knowledge will keep appearing in the near future.

The publications on Maintenance 4.0 have appeared in many scientific databases. The investigated 214 papers include 124 articles published in scientific journals, 22 book chapters, and 68 papers published in international conference materials. 

The investigated articles have appeared in 83 journals. Of the publishing titles, 50% include one article. A detailed list of journals in which the analyzed research results were published is shown in Figure 21. The journals in which at least two papers have been published are presented. The analysis includes 50 selected articles from 12 journals. 

The most significant number of publications appeared in *Economics, Management, and Financial Markets* (10 articles). Numerous publications can also be found in the *International Journal of Advanced Manufacturing Technology* (9 articles) and the *International Journal of Advanced Manufacturing Technology* (6 articles). Due to the significant diversity of topics covered in the research field under study, there were also singled-out publications on decision science (e.g., *Computers in Industry*), manufacturing engineering (e.g., *International Journal of Production Research*), or sensors (e.g., *Sensors*).

The conducted biometric analysis also concerned identifying the most frequently used keywords. The results of the conducted study are presented in Figure 22. Those keywords were included, which occurred in the articles at least two times. This limitation made it possible to identify 85 main keywords in 9 clusters. The largest cluster, “augmented reality,” contains 14 items with 53 links and a total link strength of 111, whereas the second cluster, “Industry 4.0”, encompasses 13 items, 57 links, and a total link strength of 152. This indicates the main area of research to which the publications on Maintenance 4.0 are devoted. It is worth noting the “smart factory” cluster, which has 27 links and a total link strength equal to 47. The term smart factory is mostly in relation to such keywords as big data, manufacturing, maintenance, and Industry 4.0. 

## 5. Systematic Literature Review of the Selected Papers within the Scope of the Application

The identification of the main problems and issues raised in the context of Maintenance 4.0 was based on an extensive review of the available literature. The prepared literature analysis was also supplemented by review publications in the area of Industry 4.0 or Maintenance 4.0 (e.g., works [17,38,42,80]) or reports in the field of Maintenance 4.0 and PdM 4.0 (e.g., [10,53,108]). As a result of the research carried out, five primary research areas were defined, which have been most extensively developed over the past few years. Additionally, the selected research areas appeared in review publications as those research directions that will be mainly developed in the coming years from both a research and industrial perspective (Figure 23). These are discussed in detail in the next subsections.

### 5.1. Data-Driven Decision-Making in Maintenance 4.0

The first research area under consideration is data-driven decision-making in Maintenance 4.0. In this context, Maintenance 4.0 solutions are most easily characterized in terms of the individual stages of the decision-making process (Figure 24).

Data-driven models most common in the current evolution of Maintenance 4.0 solutions are those based on statistics, pattern recognition, or artificial intelligence (AI) and those based on machine learning algorithms. The application of sensors in smart factories lies mainly in the area of control, emphasizing processes [17]. 

In this area, issues such as knowledge management (e.g., [109]), machine learning (e.g., [22,88]), artificial intelligence (e.g., [110,111,112]), IoT technologies (e.g., [113]), or big data analyses (e.g., [114]) are addressed. 

Additionally, one of the investigated research areas is data storage and processing technologies. Here, big data applications are of utmost importance. An example of a big data application for predictive maintenance is presented in [115]. The paper presents challenges encountered when building the data value chain for predictive maintenance of a grinding machine in 5G-enabled manufacturing. The data-driven value chain is also investigated by Albert in his work [116].

Another interesting problem is investigated in work [117]. The authors focus on developing and testing a data-driven condition-based maintenance tool for enabling risk-informed decision-making. The proposed approach integrates prior knowledge obtained from Preliminary Hazard Analysis–Fault Tree (PHA–FT) analysis with cyberspace defined by data-driven knowledge of system conditions. 

Additionally, the data-driven decision-making process, presented in Figure 24, is also investigated by Graham for pharmaceutical manufacturing processes based on the effective use of a process analytical technology (PAT) methodology [118]. 

At the same time, issues related to the design and implementation of so-called digital twins are currently being developed very intensively. According to the definition, a digital twin is “*a digital copy of physical assets, processes and systems with static or dynamic characteristics; often also a software term for creating virtual representations of physical systems and simulating them*” [119]. Digital Twins are a source of data that can improve the design of new products, machines, or processes. They are also used for the ongoing analysis of existing solutions to assess their capabilities, optimization, or verification of as-built documentation. With their help, it is possible to simulate optimization opportunities, conduct sensitivity analyses, or assess modification/expansion opportunities [120]. A literature review in the area of the applicability of digital twins in complex control systems was presented in the paper [121]. The digital twins’ implementation in the predictive maintenance area is reviewed in [122]. The review of applications is provided in [123].

A literature review of the area under study indicates that the basic methods used in decision-making in Industry 4.0 enterprises can be found in the works [61,124]. In addition, a significant proportion of the publications that have been selected for the literature review presented here focus on providing summaries of ongoing surveys and reports published by consulting companies such as PwC. For example, the relationships between cyber–physical systems (CPSs), AI-based decision-making algorithms, and big data-driven innovation are surveyed in [125]. In addition, the problem of robotic wireless sensor networks and real-time monitoring is studied in [126]. The survey works focus on the following: AI-based decision-making (see, e.g., [110,112,125]);Big data-driven decision-making (see, e.g., [127,128,129,130]);Business models and business processes optimization (see, e.g., [131,132]);Information systems and sensor networks for decision-making (see, e.g., [133,134,135,136]);Cyber–physical manufacturing systems (see, e.g., [136,137,138]).

### 5.2. Operator 4.0 to Support Balanced and/or Symbiotic Interaction between Humans and Machines

The history of operators’ interaction with various industrial and digital manufacturing technologies can be depicted as a so-called generational evolution (Figure 25). Operator Generation 1.0 is defined as people doing ‘manual and skilled work’ with some support from mechanical tools and manually operated machine tools. Operator Generation 2.0 represents people doing ‘assisted work’ with the support of computer tools, ranging from CAx tools to NC operating systems (e.g., CNC machine tools), as well as enterprise information systems. Operator Generation 3.0 embodies the human being engaged in ‘cooperative work’ with robots and other machines and computer tools, also known as human–robot collaboration. Operator Generation 4.0 represents the ‘operator of the future’, an intelligent and skilled operator who does ‘machine-assisted work’. It represents a new philosophy of designing and engineering adaptive manufacturing systems, emphasizing treating automation to enhance humans’ physical, sensory, and cognitive capabilities through integrating cyber–physical systems. So, we are presently focused on new cyber–physical systems with human participation—Human Cyber–Physical Systems (H-CPS), which are being designed for the following purposes [45]:Improve the ability of humans to interact dynamically with machines in the cyber and physical worlds through ‘intelligent’ human–machine interfaces, using human–computer interaction techniques designed to match the cognitive and physical needs of the operators’Improve humans’ physical, sensory, and cognitive capabilities using various enriched and enhanced technologies.

Both H-CPS goals are achieved through computational and communication techniques, similar to human-in-the-loop (H-in-the-loop) adaptive control systems [45]. 

As a consequence of the introduction of Industry 4.0 technologies, new skills are required from maintenance operators in terms of “*enhancing physical, sensory and cognitive capabilities and the ability to support major aspects of maintenance processes*” [141].

This is aimed at fostering a socially sustainable environment for production workers in the factories of the future, where “intelligent and skilled operators should not only perform ‘collaborative work’ with robots but also ‘assisted work’ by machines when needed through human cyber-physical systems, advanced human-machine interaction technologies and adaptive automation towards a ‘human-automation symbiosis’” [141].

In this context, **Maintenance Operator 4.0** refers to an operator with the ability to improve their own perception of the real world through augmented reality technology, analyzing digital data collected in collaboration with robots, with the consequent improvement of maintenance tasks in the area of their execution and control [142].

The main issues in the literature in the context of the analysis of **Maintenance Operator 4.0** mainly include:

The definition of core skills in the context of Industry 4.0 requirements (e.g., [143,144]) and operator training issues (e.g., [145,146]);

The relationship between CPS and maintenance tasks and the control role of the human (e.g., [139,147,148]);

The ways in which CPS interacts with humans (e.g., [140,149]).

Additionally, studies are devoted to knowledge management within the smart operator domain (e.g., [150,151]). In another study [152], the authors focused on new technologies implementation for Operator 4.0. They investigated the 5G-aided solutions’ influence on necessary network infrastructure for the human–machine symbiosis in the smart factory. The visual computing and simulation technologies implementation for Operator 4.0 was investigated in work [153]. The role of Operator 4.0 in the context of logistics and transportation systems was analyzed in [154] and continued in work [155], where intralogistics activities in relation to the Operator 4.0 concept are discussed via case studies. Another example may be the hybrid-augmented intelligence system developed in [156], where the intelligent digital assistant interacts with experts and operators during predictive maintenance performance.

As a result, the main challenge in this research area is how to steer the design and deployment of the Maintenance 4.0 paradigm in enterprises in the context of integrating people within CPS to achieve the desired goals. 

### 5.3. Virtual and Augmented Reality in Maintenance

Currently, the issue of the application of virtual/augmented reality solutions in the maintenance field is gaining increasing importance. Thanks to the application of augmented/virtual reality in maintenance, we can see a new perspective on the work of the maintenance employee, where it is possible to superimpose 2D and 3D documentation on the actual machine in the field, helping the technician to carry out the maintenance operation [157]. This means the online transmission of operating instructions and so-called virtual support in the process of disassembly/repair or component replacement/reassembly [158,159,160]. Using this type of solution, we can reduce the risk of individual failures/damages due to maintenance tasks carried out by a service technician with little experience. In addition, the problem of employee learning by means of trial and error is eliminated [161].

A literature review of the applicability of virtual/augmented reality in the maintenance area can be found among others in the works [17,29,158,162]. The industry’s augmented reality (AR) technologies are reviewed in [163]. The problem of integrating an AR system with an available enterprise information system was analyzed, for example, in work [164]. The overview of managerial-focused applications of AR is presented in [165]. In addition, the literature reviews focus on AR implementation in robotics (see, e.g., [166]), supply chain management (see, e.g., [167]), shipbuilding industry (see, e.g., [168]), and smart manufacturing (see, e.g., [169,170,171,172]). The AI technologies implementation possibilities in AR are reviewed in [173]. A summary of the existing knowledge on AR-powered digital twins is given in [174]. In addition, an overview of the current knowledge and future challenges of augmented reality smart glasses (ARSG) for use by industrial operators in presented in [175]. The authors focus on such categories as assembly instructions, human factors, design, support, and training. The developments in virtual reality (VR) are reviewed and summarized in [176,177], among others. 

The main application fields of VR are mostly connected with (Figure 26) [176,178]:Industrial Maintenance and Assembly (IMA);Design and Prototyping (D&P) stages of the product/system development cycle;Collaborative Virtual Environment (CVE).

In addition, the implementation of VR-based technologies should be discussed with the applied knowledge-based approach to classify them by the type of contained and presented knowledge about a product/system or a process [178]. Generally, we may distinguish three levels of knowledge of industrial VR applications [178]:General knowledge: based on interactive product/system/process visualizations. As an output, we may receive a virtual design of the investigated object;Procedural knowledge: focused on process sequence presentations and interactive machine manuals development;Applied knowledge (practical skills), which includes machine operator training simulators, “Virtual Factory” training, and ergonomics developments.

Following this, the IMA application field needs to be characterized. In this area, the research focuses on VR training in relation to knowledge transfer and an increase in the performance and accuracy of maintenance technicians [176]. A comprehensive literature review on this area is given in [158]. 

The VR-based training for a manufacturing assembly process is presented in [179]. The authors propose the integration of training simulations with virtual reality to increase assembly training effectiveness based on the “learning by doing” approach. The extension of this work is given in [180], where the authors describe a process of building a virtual training system for operators of production (assembly) workplaces in an intelligent factory. The prototype of virtual training is dedicated to use by inexperienced operators of particular stands and consists of an application (e.g., virtual environment, user interface) and specific peripheral devices. In another work [181], the authors focus on manufacturability and maintainability in the context of VR-based training implementation possibilities. They introduce Virtual Reality for the Maintainability and Assemblability Tests (VR_MATE), which encompasses VR hardware, software, and a simulation manager. In addition, two case studies demonstrate the VR training system’s ability for maintainability tests and assembly analysis. The case studies were presented for an aircraft carriage and railway coach cooling system. Another work focused on the development of VR-based training system prototype for aviation maintenance is [182]. The solution helps practitioners in the process of carrying out specific maintenance activities as removing and positioning components into aircraft structures. 

The problem of a VR-based ergonomic design process for the IMA task is investigated in [183]. The authors provide a method to quickly build a virtual IMA scenario for immersive simulation based on the traditional design platform DELMIA.

Another application area is the implementation of VR-based training for industrial robots’ proper operation and maintenance. First, the aspect of VR technologies application for machining industrial robots to improve the accuracy of teaching repetition is presented in work [184]. Next, in work [185], the authors propose using VR to train the service and maintenance of robots and robotic stations. 

In addition, the investigation of teaching design and its influence on learning performance in the operation training for CNC milling machine tools under a VR-based environment is presented in [186]. The authors analyze the sequence- (traditional) and context-based teaching designs. 

The impact of VR on learning in the context of the safety training of power producer companies is investigated in work [187]. An experiment was aimed at comparing VR training with traditional classroom training.

Simultaneously, the problem of identifying how VR training technology can be implemented in a generic operating cycle is investigated in [146]. Modular teaching is the problem studied in [188]. In addition, a methodology to develop VR tutorials and training courses for professional preparation in industrial jobs is given in [189]. An open approach to knowledge formalization and management in virtual reality applications for use in industry, especially for design and training purposes, is given in [190]. Subjective visual vertical (SVV) and subjective visual horizontal (SVH) tests are investigated in [191]. 

The second VR application field is D&P. In this area, the VR has the potential to support the design stage of the product development cycle via, among others, product testing or maintenance/manufacturing process review. In this area, a comprehensive review is presented in [192], where the authors investigate how VR supports different design functions and how they can benefit from the different degrees of immersion and additional tools. This is complemented by the work [193], where the application of VR in the assembly validation stage is presented. The authors present a developed assembly simulation to validate the feasibility of virtual assembly models. The designed system integrates low-cost, commercially available hardware to facilitate hand tracking and VR display. It is also worth mentioning [194], wherein the authors focus on the development cycle of a production system. The authors implement virtual reality (VR) into the virtual commissioning (VC) method to verify the mutual interaction of signals between the simulation in the VR environment, the digital model of the selected production system, and the control system. They focus on a real production robotic system (assembly line). 

The last application field of VR is CVE. Collaborative virtual environments are computer-enabled, distributed virtual spaces where people can meet and interact with others, with agents, and with virtual objects. In this area, we may distinguish two main research problems investigated in the reviewed papers. First, the CVE supports synchronous and asynchronous collaboration and may increase the quality of communication, knowledge sharing, and interactions among stakeholders and multidisciplinary teams. In this context, a virtual reality collaborative platform for remote teams able to work in a fully equipped environment is presented in work [195]. The problem of the utilization of virtual reality (VR) to facilitate the asynchronous collaboration of globally dispersed departments involved in the pipeline of maintenance methods and documentation creation is under investigation by the researchers in work [176]. The authors present the designed COVE-VR platform, which was developed as an academia–industry collaboration and was evaluated iteratively with subject matter experts.

Second, regarding CVE as a support for decision-making processes based on immersive VR applications, the work in [177] presents one of the investigated problems in this area. The authors review recent research on VR applications in Building Information Modeling (BIM). They discuss the current status, use cases, technologies used, and relevant future research in the field of AEC (Note: architecture, engineering, and construction)/BIM.

The use of digital twins and VR for decision-making processes in production systems is presented in [196]. The authors propose a co-simulation and communication architecture between the digital twin and virtual reality software to make optimal decisions during the design of industrial workstations. 

In another work [197], the authors introduce an architecture based on virtual reality to evaluate the civil aircraft’s maintainability. The prototype system mainly focused on accessibility during maintenance activities and was tested by importing digital 3D models of Boeing 737 and A320. 

Another aspect is the problem of VR application systems designing and development. Following this, in work [198], the authors focus on VR technologies—their aspects and limitations for supporting VR developers in creating VR industrial environments. They analyze the VR technologies based on cost, reliability, or usability criteria, among other considerations. The continuation of this problem is given in work [190], where the authors focus on a selected aspect of a methodology of building open VR systems and applications—knowledge formalization and management in industrial VR apps. In another study [199], the authors propose a distributed system monitoring tool based on VR technology. The paper discusses the design and implementation of the proposed tool and verifies the technical feasibility for further distributed system monitoring applications. 

One can also identify the works focused on both AR and VR implementation. A qualitative literature review focused on investigating the current state of these two innovative technologies and their practical application in industrial systems is presented in the works [200,201]. AR and VR are compared in [202,203,204,205]. A methodology that helps programmers to build virtual and augmented reality systems for a broad number of industrial plants is given in [206]. In addition, an overview of the opportunities for using VR and AR in a standardized value chain in various industries is presented in [207]. 

In work [208], the authors introduce the AR-based environment to support the virtual user during the assembly process. The use of AR and VR in a dynamic welding environment is given in [209]. Additive manufacturing (AM) and augmented reality technologies for supporting the workflow of producing special components are provided in work [210]. Next, in the work [211], a virtual and augmented reality in the lifecycle of semi-trailers is presented. At the exploitation stage, they support users in activities realized by operators (drivers) and service technicians. Another interesting work focuses on mixed reality applications to monitor the data of a fuel cell’s spatially resolved current density distribution [212]. In the presented case, the fuel cell represents a machine that delivers sensor values, whereas a HoloLens is the monitoring application. The use of AR with a digital twin in the area of predictive maintenance is given in work [213]. In addition, the method to visualize digital twin data by using AR technology in a real environment is presented in work [214].

Additionally, we may distinguish a growing number of research works focused on AR applications. Recent overviews of AR implementation possibilities in the industry sector and key success factors are given in [171,215,216]. The overview of AR technical components and their practical application in the industry (especially in manufacturing, maintenance, assembly, training and collaborative operations) is presented in [217,218]. Works focused on comparing traditional and AR-based learning are [219,220]. The discussion of AR technologies’ implementation possibility in decision-support processes is presented in [221].

The AR-based technologies implementation for ultrasonic pipework inspection is presented in [222]. The proposed solution is to improve the learning curve in non-destructive ultrasonic testing and to decrease the costs involved in traditional training. The discussion of AR-based technologies implementation for pipes operation and maintenance in the HVAC&R industry is given [223]. 

The AR-based solution that supports visual inspection performance on a production line is presented in [224]. The application was developed in the automotive industry. An application of acoustic processing of conditioned machine sounds and operation-related data is given in [225].

The introduction of a generic use case in the context of retrofitting and visualization of the data for real-time monitoring using a web-based AR application is given in [226]. 

The use of AR to help operators in the correct understanding of a plant and to retrieve useful information about the plant (e.g., machines layout, history of maintenance) is given in [227]. The main task is to augment the Piping and Instrumentation Diagrams (P&ID) of a selected plant/group of plants. This problem is later extended in work [227], where the authors compare the AR application with the currently applied practice, based on paper documentation, for an information retrieval task within a maintenance procedure.

The analysis of possibilities of the practical application of modern AR solutions in the industry, with a particular focus on remote support for maintenance operations and training of production employees, is given in [228]. Two experiments are described to determine the impact of various environmental conditions on the possibility of using AR Remote Support.

A solution for remote maintenance based on off-the-shelf mobile and AR technologies is provided in [229]. The proposed application allows for the remote connection of a skilled operator in a control room with an unskilled one located where the maintenance task has to be performed. The AR application possibilities to support remote maintenance as a service in the robotics industry are given in [230]. The main objective of this paper is to develop an internet-based, service-oriented system that implements AR technology for enabling tele-maintenance by the cooperation of the end user and the manufacturer. In another work [231], the authors focus on AR-based remote maintenance processes supported by cutting-edge optical head-mounted display technology. The proposed solution is based on the remote connection between a maintenance worker who wears the HoloLens and an expert who observes the live-stream video on a laptop. The expert provides advice orally or via 3D virtual annotations that are transferred back to the maintenance worker. 

Examples of the use of AR include bus maintenance systems (e.g., [232]), power plant maintenance (e.g., [233]), railway systems (e.g., [234]), and military systems (e.g., [235]). The concept of a system for assisting diagnostic processes using AR technology was presented among others in works [236,237]. An example of a platform used in the training process of service engineers based on virtual reality is shown in work [238]. The AR application concerning workforce skills is given in [239].

In addition, other AR-based application areas include education engineering (e.g., [240,241]), manufacturing processes performance (e.g., [242,243,244,245]), assembly workstations support (e.g., [246,247,248,249]), flexible manufacturing systems layout planning (e.g., [250]), product inspection during the design process (e.g., [251]), learning and training of 3D printing process (e.g., [252]), construction industry (e.g., [253,254]), AR application selection framework process (e.g., [255]), and logistics operations (e.g., [256]).

Finally, worth mentioning here is also that such systems have recently become very expensive, and there is the problem of battery durability (limiting the use of solutions in practice). In addition, the use of such a solution requires the company to invest in a machine diagnostic system, build a 3D model of the machine, develop and introduce assembly/disassembly instructions and all the information about individual machine components so that they are available to the service technician during maintenance [159]. 

### 5.4. Maintenance System Architecture

In the given research area, the authors of the publication focus primarily on the problem of designing cyber–physical systems, taking into account the essential elements and potential organizational and technological solutions that allow efficient and economical operation in Industry 4.0 organizations. CPS is defined as integrating computational and physical capabilities that can interact with humans through many new modalities [257]. 

Problems are considered in the context of both vertical and horizontal integration. At the same time, cyber–physical systems and their architecture include elements such as (Figure 27) [257]:Sensor network;Real-time data transmission;Data centers;Control centers;Control systems;System users.

Source: own contribution based on [257].

CPS is characterized by close interconnection and coordination between networked and physical systems. Through the integration and in-depth collaboration of computing, communication and control (3C) technologies, CPS can provide real-time sensing, control and information services to large engineering systems [84]. This problem is particularly developed in the works [84,257,258,259]. Additionally, in work [260], the authors review and summarize the differences between CPS’s traditional centralized and hierarchical architectures, which rely on decentralized decision-making and control. The concurrency and synchronization problem in CPS is reviewed in [261]. A short survey regarding the concept, architecture, and challenges for deploying cyber–physical systems within the concept of Industry 4.0 is then presented in [262]. The problem of knowledge-based systems and their architecture is also overviewed in the work [263]. 

The problem of CPS architecture designing based on the Lean Six Sigma approach is reviewed in work [264]. The 8 C architecture, i.e., connection, conversion, cyber, cognition, coalition, customer, and content, is applied for conducted review.

Another problem in this area is connected to IoT system architecture. This aspect of CPS is investigated in works [265,266,267,268,269,270]. In work [266], the authors review four main IoT architectures, namely (a) IoT identity management architecture, (b) IoT edge computing solution architecture, (c) IoT information modeling and management architecture, and (d) autonomous industrial IoT communication architecture. They also define the main limitations of the existing studies. 

The IoT systems architecture in the context of digital twins designing is investigated in work [267]. The main architectural aspects, such as internal structure or runtime environment, are discussed. 

The IoT architecture security is also one of the important aspects being investigated recently. In work [265], the authors focus on security-based IoT architecture analysis. In addition, the authors in [269] review IoT ecosystems and the possibilities of their security improvement based on new technologies (e.g., blockchain, AI, ML) implementation. 

The application of IoT architecture is presented in [270], where the authors focus on an organic rankle cycle turbine. Another work [268] summarizes the relations between CPS and IoT. 

The next problem regarding the CPS architecture is connected with the requirements for cloud computing implementation. The cloud-based CPS are investigated in works [271,272,273,274,275]. In work [271], the authors focus on cloud-based applications in the context of data sharing and computing resource availability. The new federated cloud computing model is presented. In another work [272], service-oriented architecture is introduced. The proposed service architecture attempts to cover the basic needs for monitoring, management, data handling, and integration by considering the disruptive technologies and concepts that could empower future industrial systems. The cloud-based industrial automation system architecture is, in turn, introduced in work [273]; the author bases on networked control systems (NCS), industrial control theory, and computing theory implementation. The last paper in this area focuses on developing software system architecture based on model-based system engineering (MBSE) and cloud computing. This system architecture’s building process follows the ISO/IEC/IEEE 42010 and federal enterprise architecture framework. An interesting view of cloud-based maintenance services is presented in [275]. The main problem here is properly managing maintenance data concerning stakeholder groups. 

Another investigated problem is connected with big data and the ML approach implementation for CPS architecture. The data-driven issues in CPS are investigated in works [276,277,278]. A big data architecture for Industry 4.0 is introduced in work [278]. The authors describe its main layers and components, supporting data collection, integration, storage, processing, analysis, and distribution. This problem is later extended in work [277], where the authors analyze the implications of the inclusion of ML components into traditional anomaly detection systems in relation to IT system architecture. In the last work [276], a data-based SAE–SVM (Serious Adverse Event-Support Vector Machine) approach is proposed to diagnose transmission faults for multi-joint industrial robots. The proposed solution is based on a deep learning approach implementation.

Moreover, recent developments focus on AI and virtual reality implementation (see works [279,280,281]). Here, the investigated issues regard:Designing a novel cognitive architecture for artificial intelligence in cyber–physical production systems [279];A six-layer digital twin architecture with upstream and feedback flows [280];Designing a middleware system architecture that can automate X-Reality (XR) systems configuration and create tangible in-site visualizations and interactions with industrial assets [281].

Worth noting are also works focused on special reference architecture models’ presentation and implementation. Here, such system architecture models and approaches like Reference Architecture Model Industry 4.0 (RAMI 4.0) or Industrial Internet Reference Architecture (IIRA) are introduced. 

The RAMI 4.0 model’s use and implementation possibilities are presented in works [282,283]. First, in work [282], the authors introduce a domain-specific systems engineering approach using a Domain Specific Language (DSL) based on the results of this reference architecture. Later, in work [283], the authors review multi-agent systems design patterns based on, among others, RAMI 4.0 requirements. The IIRA (Industrial Internet Reference Architecture) model is investigated in work [284], where the author focuses on autonomic CPS for Industry 4.0. 

The published works also address issues related to the safety or reliability of the designed architecture systems. The safety aspects of CPS architecture are analyzed in [285] in the context of dynamic safety assurance for Industry 4.0 technologies. In another work [286], the resilience aspect of CPS architecture is investigated. The authors propose a design method for a resilient architecture of a cyber–physical production system that can deal with disturbances and failures in a discrete-event process. The reliability issues of CPS architecture are investigated in [287]. The authors focus on a communication-based train control system and its reliability evaluation. 

The CPS architecture implemented in maintenance is defined as another research problem investigated in the reviewed literature. The main aspects regard: the implementation of a reference architecture for cyber–physical systems (CPS) to support the condition-based maintenance (CBM) of industrial assets [288], the designing of industrial control system architecture for large-scale industrial and infrastructure construction projects [289], self-aware machines development and support [290,291], predictive maintenance system architecture [292,293], manufacturing process scheduling and control [294], an industrial event-driven architecture implementation [295], and key performance indicators for CPS development [296].

Additionally, few works focus on some special applications of CPS with an investigation of system architecture. For example, a warehouse management system architecture based on IoT implementation is presented in [297]. The implementation possibilities are described for a textile company. Reconfigurable manufacturing supply (RMS) chain architecture for ensuring rapid and autonomous reconfiguration of production systems while considering unpredictable supply-chain factors and their impacts on production capacity and operational/energy cost is investigated in work [298]. It is also worth highlighting the work [299], where the authors presented the concept of Smart Innovation Engineering (SIE) that enhances the product innovation process in a manufacturing company. 

### 5.5. Cybersecurity in Maintenance

Cybersecurity is becoming a key aspect of operational technology (OT) in the context of digitalization and the development of industrial networks. This is confirmed by a number of studies and reports dedicated to the topic and published in recent years (e.g., [300,301,302,303]). A brief overview of the main Industry 4.0 technologies and paradigms concerning cybersecurity aspects is given in the work [304]. 

According to the Industrial Cybersecurity Centre (CCI) [305], industrial cybersecurity is “*a set of practices, processes and technologies designed to manage cyber risks arising from the operation, processing, storage and transmission of information used in industrial organizations and infrastructures, using a people, process and technology perspective*”. 

The stages of development of cybercrime and the most significant cyber-attacks that have taken place in recent years are identified in work [306]. Thus, the cybersecurity strategies built in a company should be fully integrated from the outset with its organizational strategy and information technology and provide secure, proactive, and resilient IT solutions. Therefore, these strategies are currently built at three key levels: device, industrial network, and device security management (Figure 28). The impetus for the development of solutions related to cyber risk in business was provided by, among others, the EU Cyber Security Directives (NIS and NIS2) [307,308] and the Polish Act on the National Cyber Security System (KSC) [309]. Specific standards have been defined in some industries, such as the VDA ISA/TISAX for the automotive industry. On the management level, on the other hand, international standards such as ISO 27001 [310] or ITIL/ISO 20000 [311] are used. 

In addition, every country develops its cyber security strategy to define and institutionalize the national cyber security system. This strategy covers the national documents explaining what should be performed to achieve a high common level of network and information systems security. The cybersecurity strategies developed in the selected EU countries and NATO members are reviewed in work [312].

According to CGI [300], countering cyber threats requires a holistic approach based on three basic steps: (1) identify and evaluate (assess), (2) monitor, and (3) control and secure (Figure 29). These steps are designed to help organizations achieve a mature level of cyber security, secure their most valuable assets, and ensure business continuity. 

Following this, the key issues currently addressed in the literature include:Cyber risk analysis and assessment (cyber risk evaluation) (e.g., [314,315,316,317]);Cyber vulnerability assessment (e.g., [318,319,320,321]);Resilience to cyber-attacks (e.g., [322]);Safety aspects of digitalized CPS (e.g., [323,324,325]).

The problem of cybersecurity in Maintenance 4.0 is also investigated concerning the main requirements for information security from an engineering perspective [326]. The author focuses on four engineering-related requirements for achieving security: security requirements elicitation, security analysis, security design, and security validation. The systems engineering approach for cybersecurity analysis is also used in work [327]. The authors determine the body of knowledge for creating a postgraduate cybersecurity module.

The reviewed literature also focuses on the primary Industry 4.0 technologies and their cybersecurity [328,329,330,331,332,333,334]. An overview of the leading enabling technologies in Industry 4.0 and possible cybersecurity threats to them is presented in [331]. Key security issues related to the implementation of IoT are investigated in works [328,329,333]. Cloud computing security and big data are analyzed in [330,332,334]. 

The decision-making support system developed for system analysis and searching of optimal versions for cyber security facilities placement of an enterprise or organization distributed computational network (DCN) is given in [335]. The data-driven manufacturing processes and their cybersecurity is investigated in [336]. The manufacturing information architecture and its security is discussed. This problem is also extended in work [337], where interoperability services are investigated. The requirements of cybersecurity information exchange are defined. 

The problem of human factor security in IoT systems was deemed worthy of investigation and analysis in work [320]. The authors focus on socio-technical dimensions in relation to industrial environment cyber security improvement possibilities.

Additionally, few works can be identified focused on cybersecurity for specific systems. For example, in work [338], the authors investigated aviation cybersecurity. They overview known frameworks to determine the current maturity status at the international, regional (the European Union), and national (the Republic of Poland) levels. In addition, in work [339], the authors address the cybersecurity issues of autonomous haulage systems (AHS) in the mining industry. They investigate AHS cybersecurity in relation to communication challenges, cybersecurity, and safety. Robotics cybersecurity is reviewed in work [319]. The authors focus on the main security vulnerabilities, threats, risks with their impacts, and the main security attacks within the robotics domain. The robotic/mechatronic systems cybersecurity is also reviewed in work [322]. The authors focus on the problem regarding the possibility of cyber-attack detection. For this purpose, they developed a laboratory stand devoted to the design of Industry 4.0 technologies. The extension of this work is given in [340], where the authors focus on detecting anomalous behavior in cyber–physical devices caused by threat models based on Stuxnet-like and BlackEnergy-like malware. 

The safety aspects of digitalized offshore oil and gas production systems are reviewed in work [323]. In the work, five principal attributes related to the cybersecurity of safety instrumented systems (SIS) are investigated and carefully reviewed, namely: governing standards and regulatory frameworks, risk intelligence, barrier design, continuous revision and management, change control, surveillance, and system resilience, and industry sector-specific cybersecurity culture. The impact of cybersecurity attacks on power systems is investigated in [341].

In addition, supply chain cybersecurity is reviewed in work [342]. The authors focus on such aspects as network security, information security, web application security, and IoT. E-governance and its cybersecurity is the subject of interest to the authors of [343]. 

For a summary of cybersecurity issues, see the article [344]. The cybersecurity perspectives for AR and digital twins are reviewed in the work [174], whereas AI aspects are summarized in work [345]. Machine learning and cybersecurity aspects are reviewed in work [346].

## 6. Discussion

The main goal of this article is to provide a complete review of the existing literature to present an up-to-date and content-relevant analysis in the area of Maintenance 4.0 main application fields. Two hundred fourteen articles that satisfied the defined selection criteria were compiled, which allows answering the stated research questions. 

RQ1 intended to discover the leading trends in Maintenance 4.0 approaches and investigate their evolution over the last decade. The main research outputs here are discussed broadly in Section 5.1, Section 5.2, Section 5.3, Section 5.4 and Section 5.5. 

In the defined five application areas, the scope of issues covered is very complex, ranging from the presentation of technological solutions dedicated to predictive maintenance/Maintenance 4.0 to issues related to the analysis of acquired data and the need to make complex operational decisions. There are a number of papers concerning topics that are primarily related to the key terms of sensors, smart factories, deep machine learning, the Internet of Things, or big data analytics. The smart factories’ maintenance processes are mostly based on digitalization, data-driven manufacturing, and digital twins. According to the literature review, the application field that has been most extensively developed in recent years is VR-based and AR-based technologies in maintenance. Over the last ten years, we have selected 96 publications devoted to the problems of VR/AR-based training systems development, VR/AR-supported production development cycle processes, or Collaborative Virtual Environment areas. On the one hand, a lot of work aimed at dedicated solutions for production and maintenance process support for various industry sectors was highlighted. On the other hand, the clear industrial demand for this type of solution indicates that this research area will continue to be widely developed in the future in the context of technological (hardware) or software solutions.

The conducted systematic analysis of the selected literature makes it possible to answer the second research question. 

RQ2 intended to discover the future research directions and perspectives in Maintenance 4.0 selected application fields. 

One of the main identified knowledge gaps is the automation of diagnostic procedures and methods in the context of large amounts of data generation. This process drives the need for the development of more complex and sophisticated methods of autonomous analysis of condition-monitoring data. As a result, the big data analysis methods supported by various AI-based solutions will be one of the main challenges in both scientific and practical terms. Another issue in this context is connected with the operators’ skills necessary to realize the full potential within the gathered data. In addition, the problem of data acquisition concerning the required infrastructure to record and transmit data to maintenance engineers will also constitute one of the biggest challenges of today’s industry sectors. The last challenge in this area is uncertainty at various maintenance decision-making stages. Here, the detection stage (signal processing), diagnostics, and prognostics are strictly affected by epistemic and ontological uncertainty regarding, among others, the operational environment, measurement errors, and modeling of degradation. In addition, the maintenance decision-making process should also be investigated regarding the availability and (long-term) reliability of sensors in manufacturing plants. Not every plant has the same data capture and storage capabilities. 

At the same time, a key issue is properly selecting intelligent sensors for smart factories in designing proper predictive maintenance systems that allow for efficient decision-making processes. There is a wide range of available sensors in the market, and the appropriate decision-making process based on adequately defined criteria is one of the knowledge gaps. The well-known approaches are mainly based on four criteria: cost, flexibility, size, and sensitivity. 

In the context of the identification of the leading research gaps, the issues of knowledge-based solutions and data-driven techniques will constitute one of the future research development directions in the area of PdM 4.0. Here, one of the knowledge gaps is the integration of data-driven techniques with VR/AR-based solutions and data gathering and analysis issues with the support of cloud-related technologies. Moreover, despite the growing interest in machine learning techniques and digital twins’ application in predictive maintenance, there is still room for development in anomaly detection (abnormal patterns and events). 

Another research gap may be connected with developing new performance models in maintenance focused on the new concept of “Machine as a Service”. This concept uses connectivity with industrial IoT touchpoints to deliver information about real-time operations to extend the machine’s capabilities to meet business-wide goals. A new concept will be developed in the future in the context of maintenance management, product optimization opportunities, or new business models definition. 

Finally, based on the presented overview, it can be concluded that these issues will evolve both in the literature and in practice towards providing more precise and faster diagnostic systems, allowing companies to take the right action in even shorter decision-making times. In this context, one will see the rise of augmented and virtual reality-based solutions as a new form of communication between operators and maintenance engineers. Devices such as smart glasses allow engineers to create a digital representation of faults/failures and diagnose machine conditions with greater detail and reliability. Digitization and the increasing use of big data technologies will provide greater connectivity between machines and equipment and allow cost-effective decision-making. Ensuring an adequate level of cybersecurity in the modern smart factories of the future will remain a fundamental challenge. 

At the same time, when looking at Industry 4.0 and Maintenance 4.0, it is also essential to consider the new challenges facing companies. In 2021, the European Commission officially presented a report: *Industry 5.0: Towards a sustainable, human-centric and resilient European industry* [347], introducing the new concept of Industry 5.0. According to the report, Industry 5.0 complements the existing concept of Industry 4.0 by emphasizing the importance of research and innovation as drivers of the transition to a sustainable, human-centered, and resilient European industry (Figure 30). A review of these three leading characteristics of Industry 5.0 was provided in [348]. A preliminary survey-based tutorial on potential applications and supporting technologies of Industry 5.0 is presented in [349]. The first attempt to investigate the relations between human-in-the-loop-based maintenance framework and physical asset resilience can be found in [350].

## 7. Conclusions 

This paper presented the performed meta-analysis and systematic review that addresses the issues of Maintenance 4.0 in the context of the main application areas. The discussion of specific application areas is provided through the analysis of 214 recent papers and a macro-view of publication trends in the overall state-of-the-art. In addition, the main research and knowledge gaps are discussed to identify the main trends and challenges facing the Maintenance 4.0 technologies development. 

The presented work suffers several limitations, mostly related to the reviewing methodology assumptions connected with publication collection, searching strategy, and filtering criteria. The search strategy is biased by the problematic key term “predictive maintenance”, which was omitted in the search engine. The term predictive maintenance is not interchangeable with “Maintenance 4.0” in the same meaning. In addition, the literature related to medicine, health issues, or environmental aspects is omitted in the conducted overview analysis. The authors focused on manufacturing and production-related publications only. Moreover, the conducted literature analysis does not consider the quality of the investigated publications based on times cited. The authors present only the most cited keywords in their bibliometric analyses. 

Moreover, it is worth noting here that topics concerning Industry 4.0 in maintenance have rapidly evolved, especially in particular industry sectors. The presented literature overview provides a general macro-view without considering the specific characteristics of individual industrial sectors. In this regard, it should be noted that the development of the Maintenance 4.0 concept in various industrial sectors, such as the mining industry, is characterized by its dynamics and the complexity of the proposed solutions and research trends. 

The authors recommend that a more exhaustive literature review, especially related to domain-specific areas, should be performed for future work. Furthermore, topics such as sustainable maintenance systems or the mitigation of environmental impacts related to new technologies implementation may require further exploration. Another interesting research direction may be connected with new business model development. In this area, such concepts as “Machine as a Service”, “Software as a Service”, “Software as a Service” are other trends suitable for future research.

## Figures and Tables

**Figure 1 sensors-23-01409-f001:**
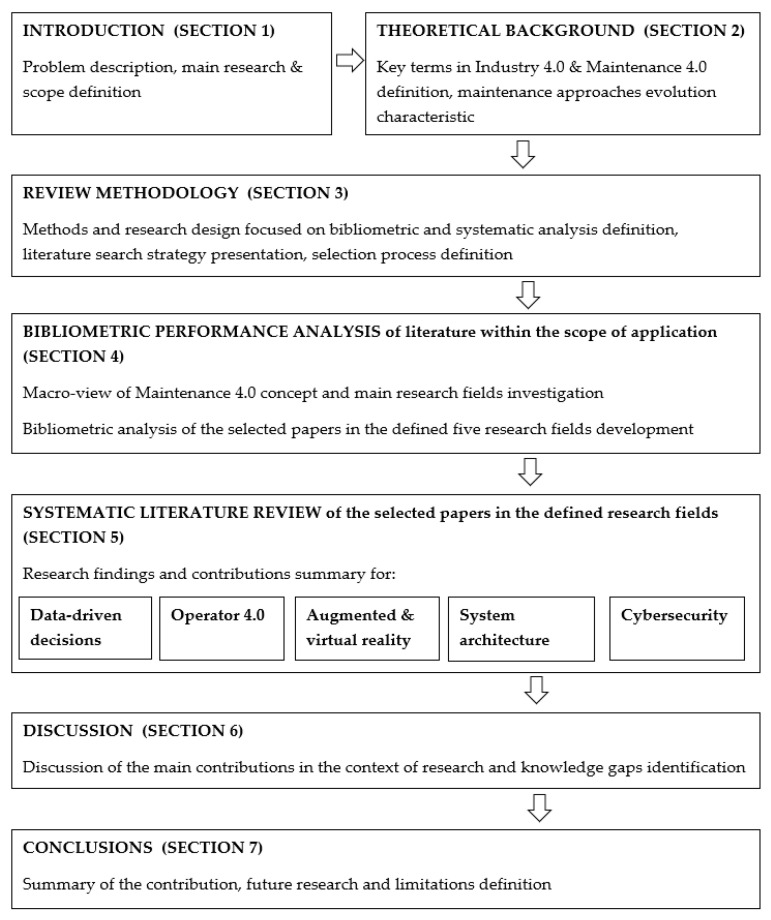
The structure of the article. Source: own contribution.

**Figure 2 sensors-23-01409-f002:**
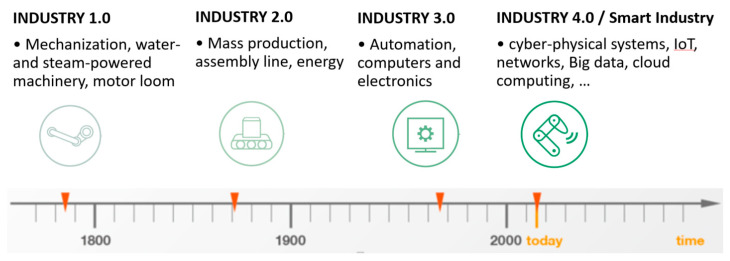
Fundamental trends of the industrial revolution. Source: own contribution based on [44,45].

**Figure 4 sensors-23-01409-f004:**
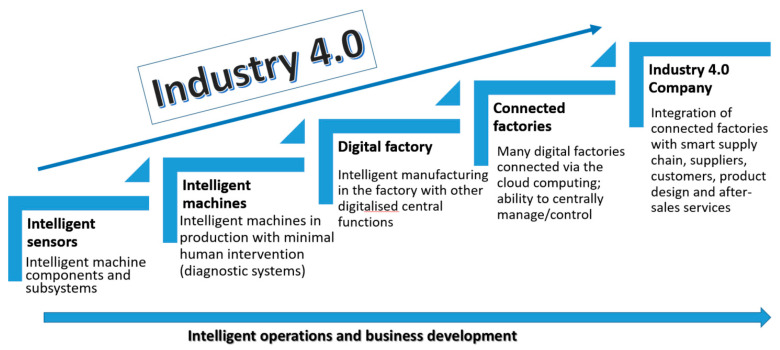
Development directions in Industry 4.0—what are businesses working towards? Source: own contribution based on [52].

**Figure 5 sensors-23-01409-f005:**
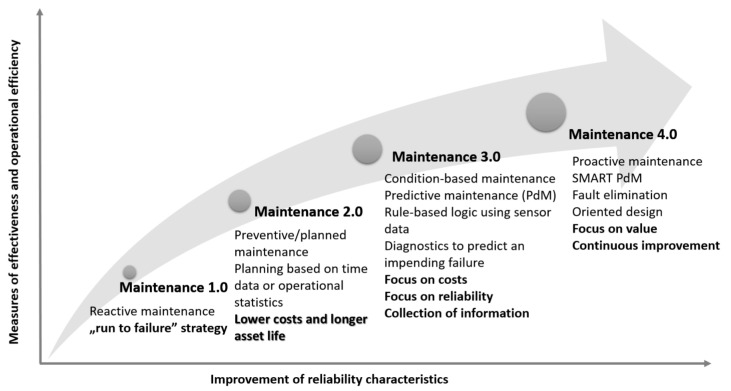
Main approaches to maintenance. Source: own contribution based on [1].

**Figure 6 sensors-23-01409-f006:**
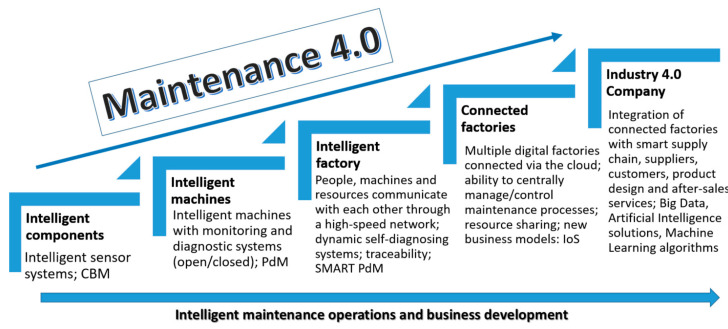
Development directions in Maintenance 4.0. Source: own contribution based on [52].

**Figure 7 sensors-23-01409-f007:**
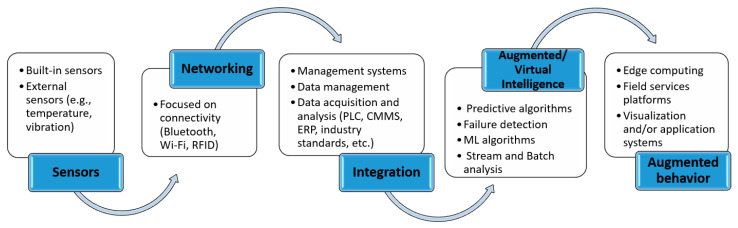
Technologies enabling PdM processes in the context of Industry 4.0. Source: own contribution based on [53,86,87].

**Figure 8 sensors-23-01409-f008:**
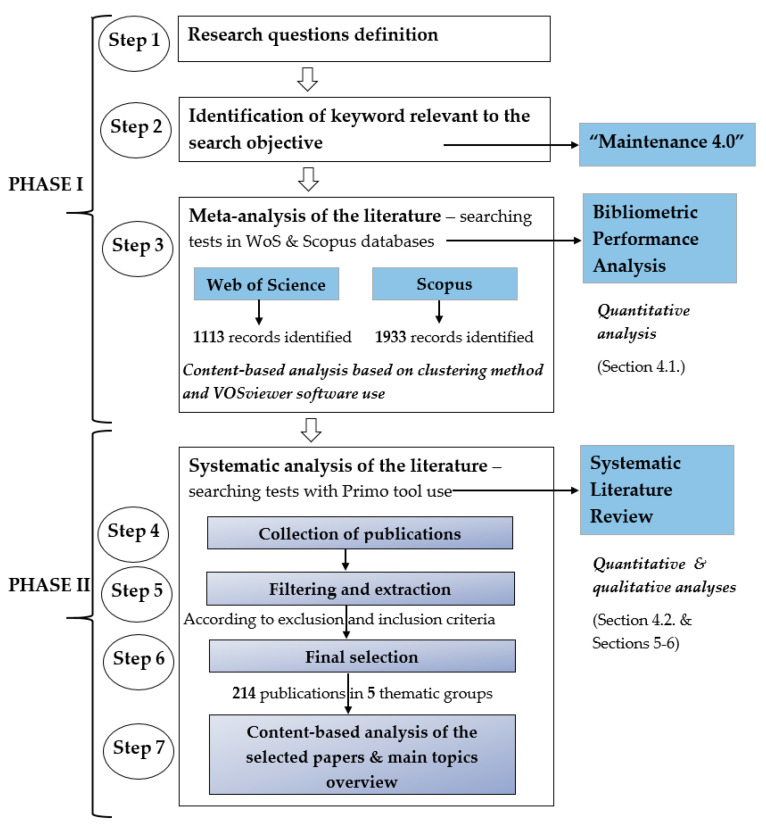
Research framework and methods/tools used. Source: own contribution.

**Figure 9 sensors-23-01409-f009:**
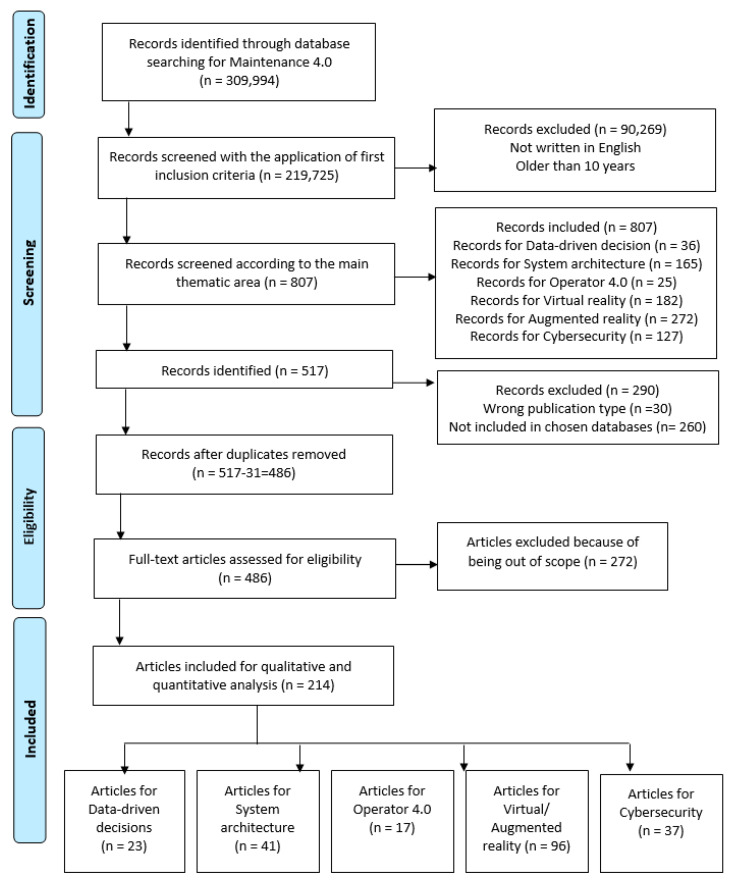
PRISMA-based flowchart of the systematic selection of the relevant studies in the analyzed research area. Source: own contribution based on [93].

**Figure 10 sensors-23-01409-f010:**
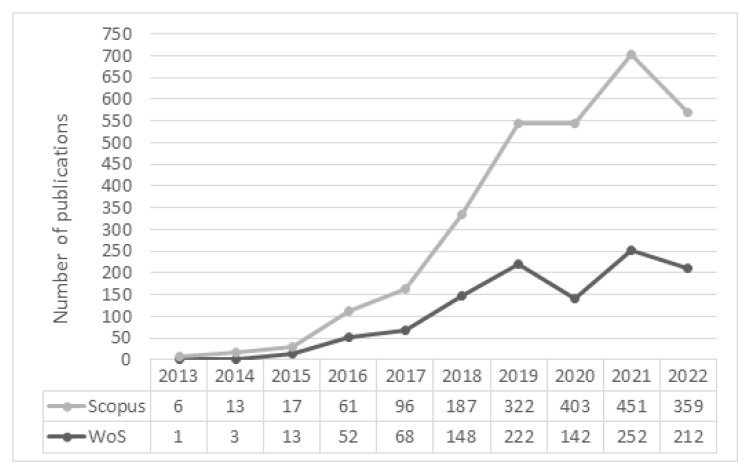
Publications in engineering disciplines from 2013–2022 that included the term Maintenance 4.0 and were published in Scopus and Web of Science databases.

**Figure 11 sensors-23-01409-f011:**
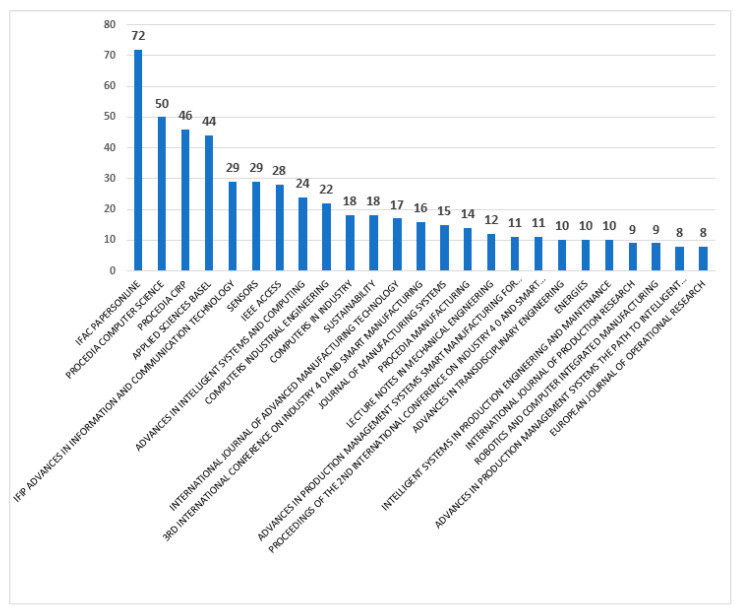
Publication titles in which the Maintenance 4.0 research area papers were the most frequently published (for the Web of Science database).

**Figure 12 sensors-23-01409-f012:**
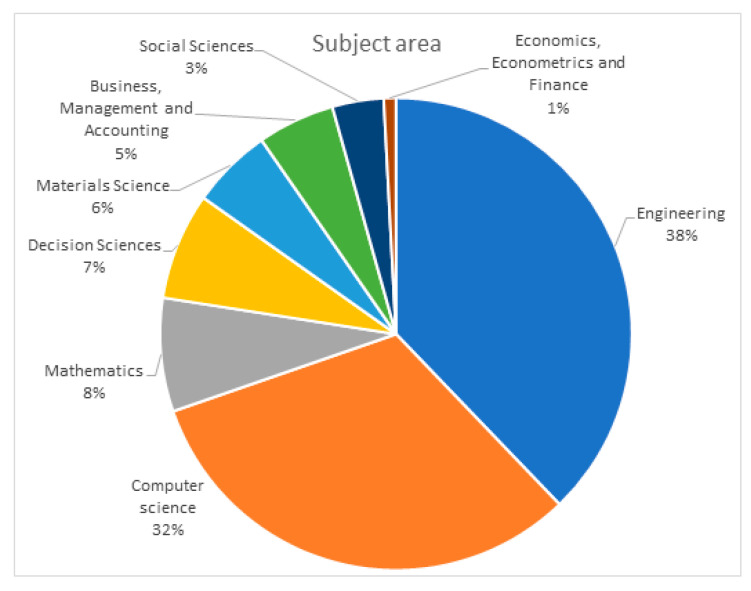
Subject areas in which Maintenance 4.0 papers were the most frequently published (for the Scopus database).

**Figure 13 sensors-23-01409-f013:**
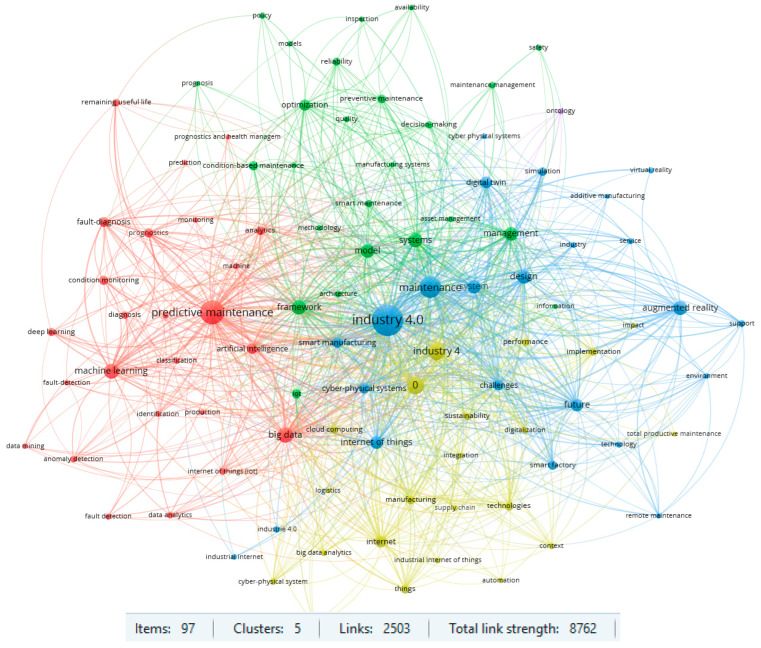
Analysis of keyword strength for the analyzed 1113 publications from the WoS database. Source: own development using VOSviewer software [101].

**Figure 14 sensors-23-01409-f014:**
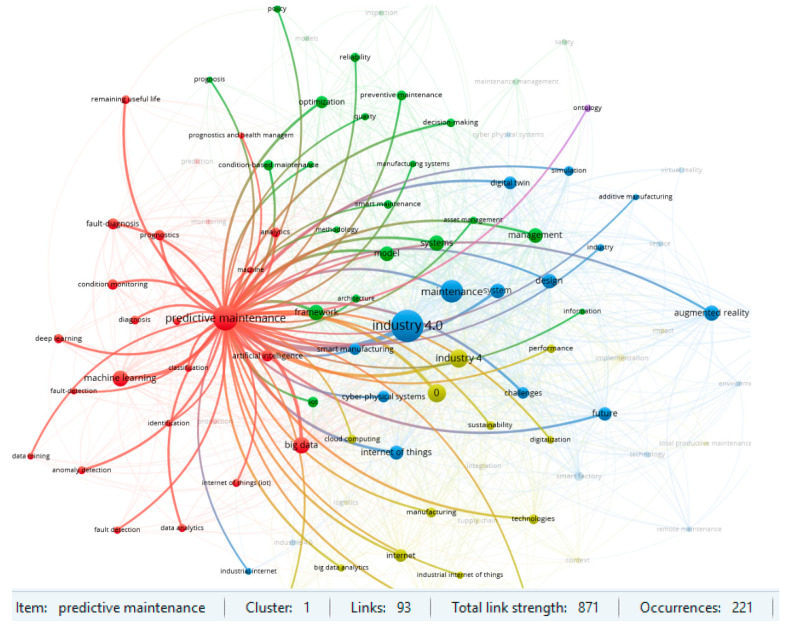
Analysis of keyword strength for the first cluster of publications from the WoS database. Source: own development using VOSviewer software [101].

**Figure 15 sensors-23-01409-f015:**
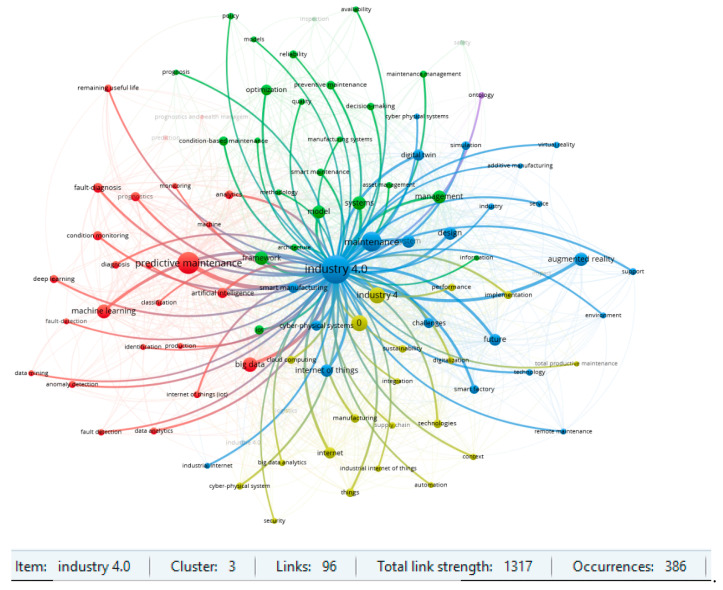
Analysis of keyword strength for the third cluster of publications from the WoS database. Source: own development using VOSviewer software [101].

**Figure 16 sensors-23-01409-f016:**
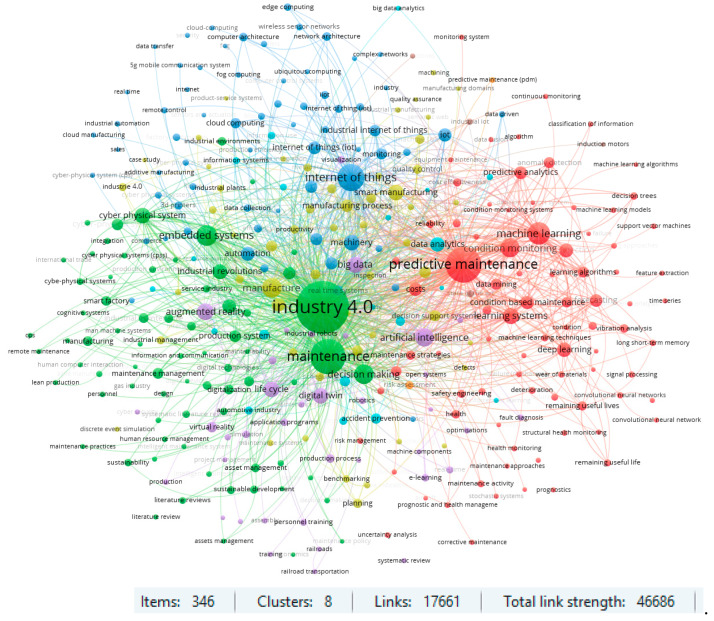
Analysis of keyword strength for the analyzed 11933 publications from the Scopus database. Source: Own development using VOSviewer software [101].

**Figure 17 sensors-23-01409-f017:**
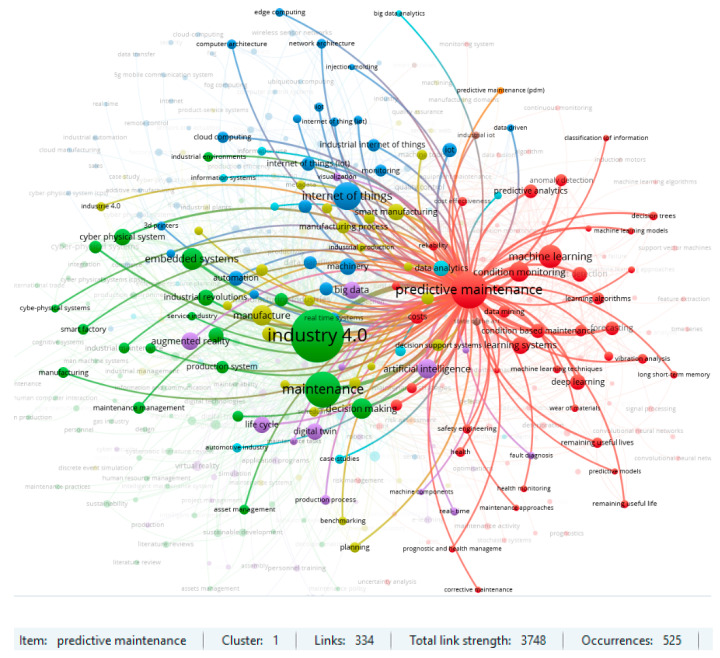
Analysis of keyword strength for the first cluster of publications from the Scopus database. Source: own development using VOSviewer software [101].

**Figure 18 sensors-23-01409-f018:**
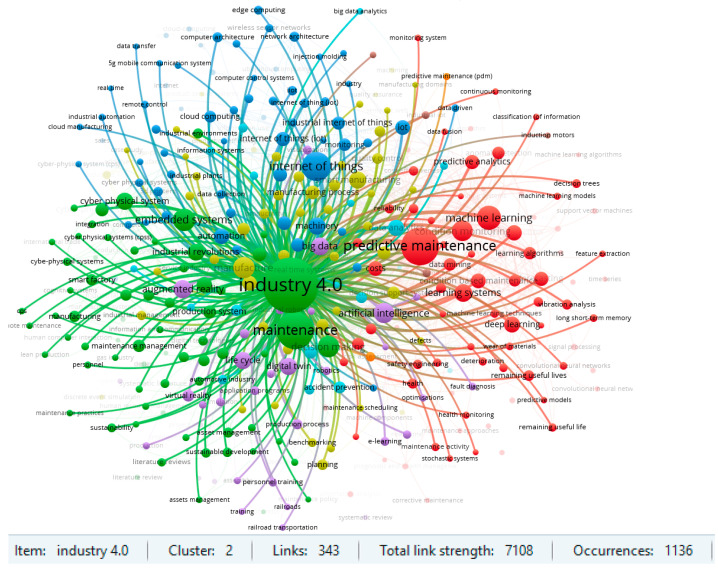
Analysis of keyword strength for the second cluster of publications from the Scopus database. Source: own development using VOSviewer software [101].

**Figure 19 sensors-23-01409-f019:**
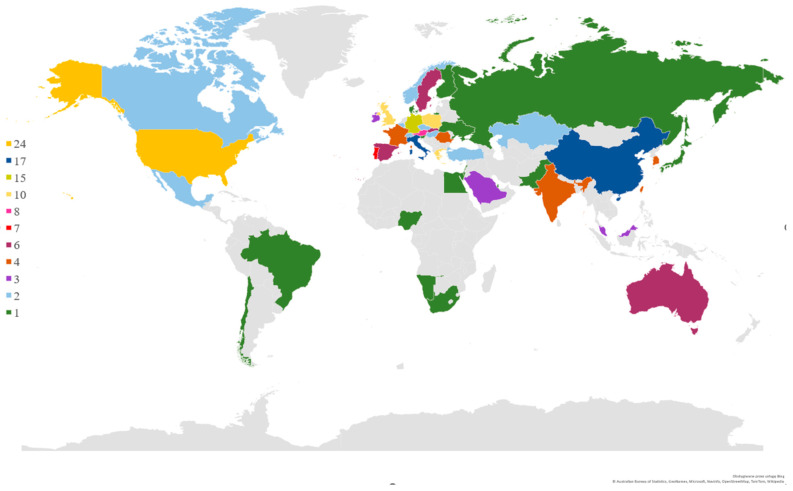
A number of papers by the location where the investigated study took place.

**Figure 20 sensors-23-01409-f020:**
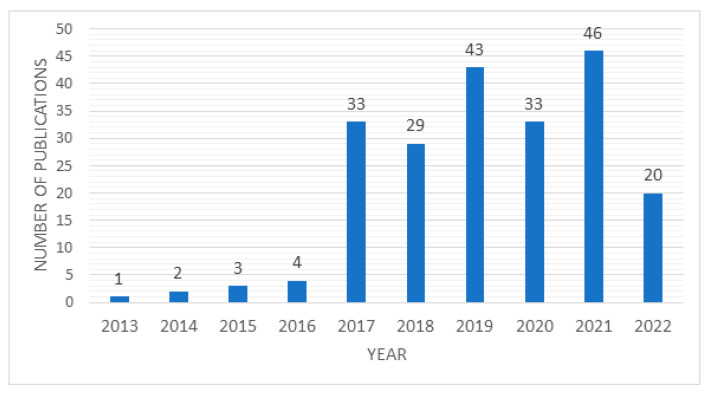
A number of papers by publication year.

**Figure 21 sensors-23-01409-f021:**
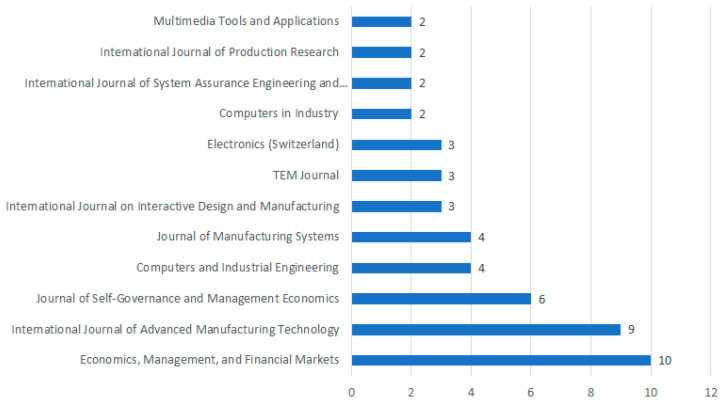
A number of papers in each investigated journal (for journals with at least two published papers from the analyzed 214 articles).

**Figure 22 sensors-23-01409-f022:**
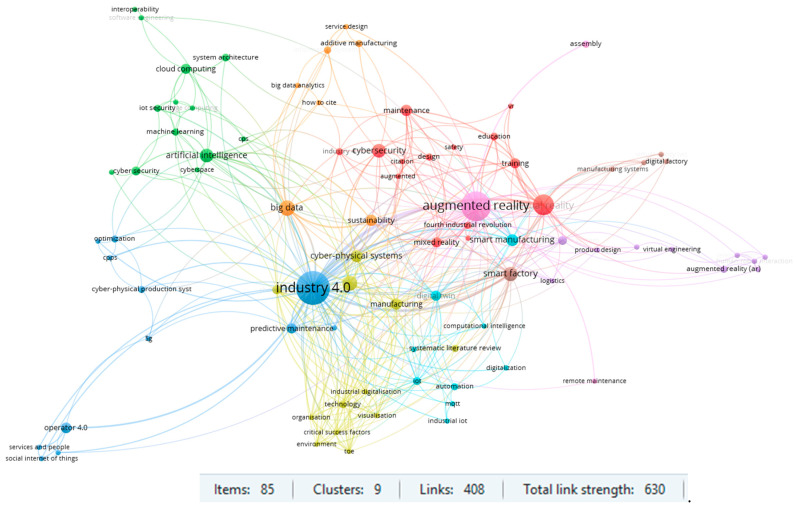
A number of the most frequently occurring keywords in the analyzed papers. Source: own development using VOSviewer software [101].

**Figure 23 sensors-23-01409-f023:**
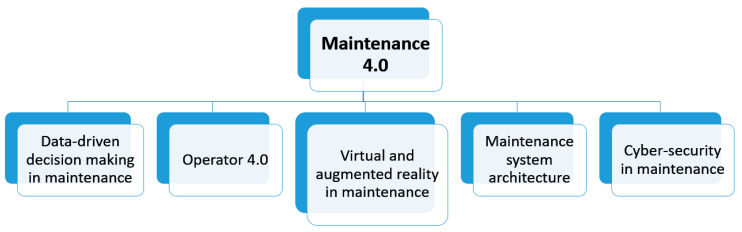
Main issues addressed in the context of the Maintenance 4.0 concept.

**Figure 24 sensors-23-01409-f024:**
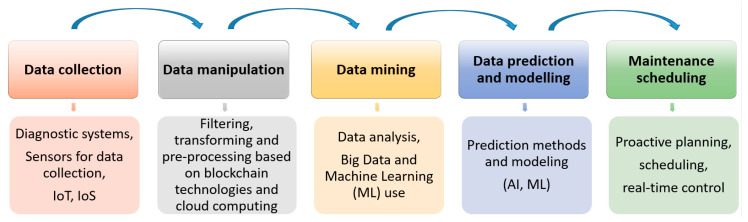
Data-driven decision-making in maintenance. Source: own contribution based on [17].

**Figure 25 sensors-23-01409-f025:**
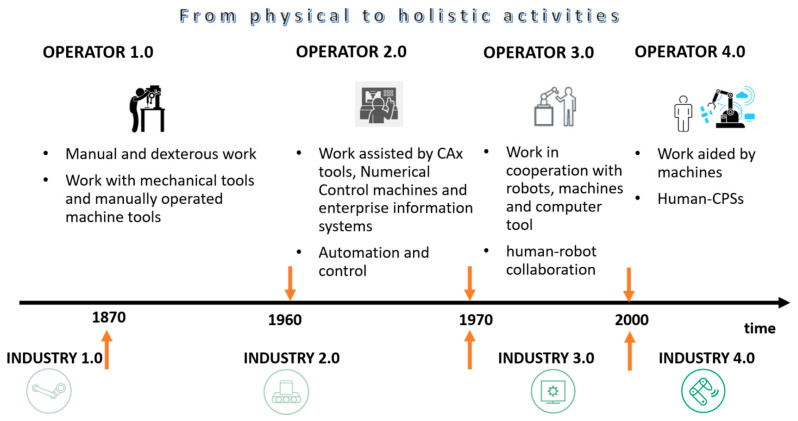
(R) Evolution of the Operator Concept Generation. Source: own contribution based on [45,139,140].

**Figure 26 sensors-23-01409-f026:**
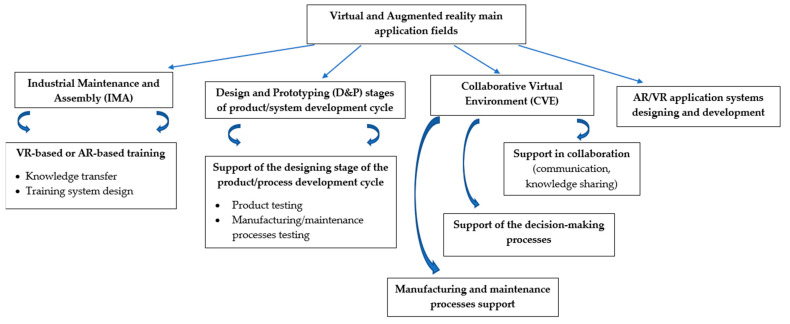
Virtual and Augmented reality main application fields. Source: own contribution.

**Figure 27 sensors-23-01409-f027:**
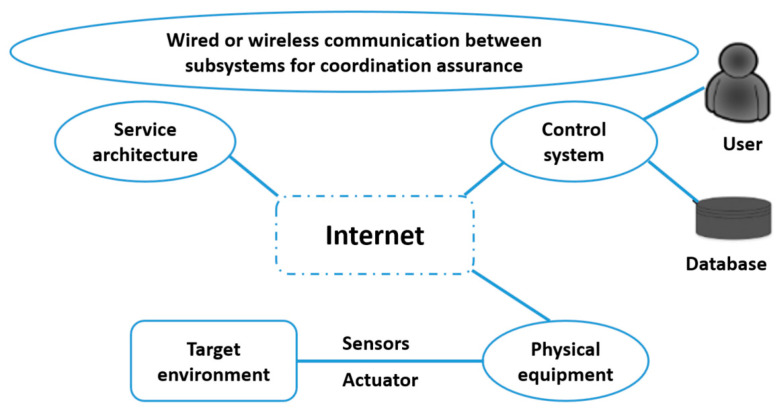
Service-oriented CPS architecture.

**Figure 28 sensors-23-01409-f028:**
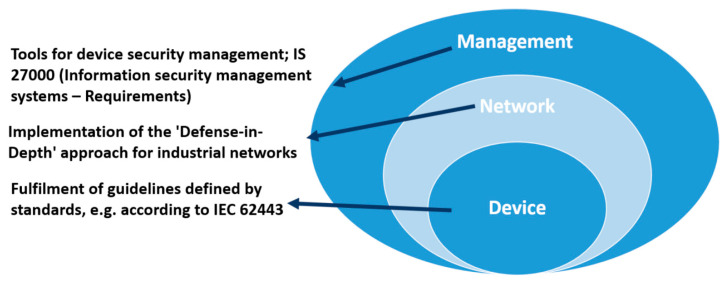
Key levels of cyber security and users’ requirements and needs. Source: own contribution based on [313].

**Figure 29 sensors-23-01409-f029:**
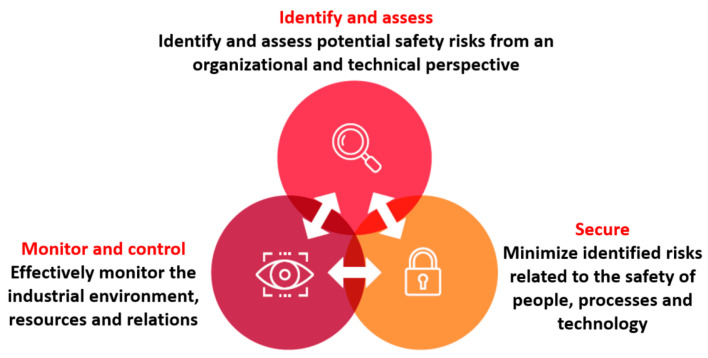
Cyber security in IT and OT. Source: own contribution based on [300].

**Figure 30 sensors-23-01409-f030:**
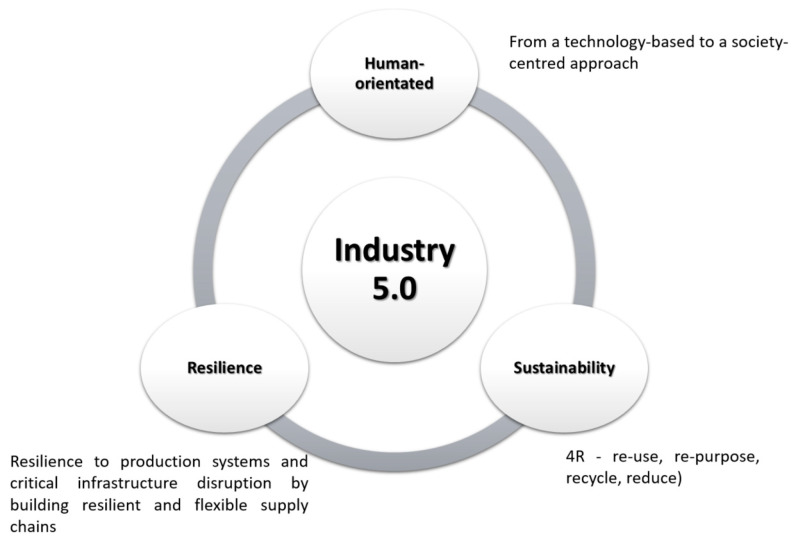
The concept of Industry 5.0. Source: own contribution based on [347].

**Table 1 sensors-23-01409-t001:** A summary of the recent papers focused on the literature overview in the area of Maintenance 4.0.

Ref.	Publication Year	Research Objectives	Methodology Used	Databases Analyzed	Papers Analyzed	Focused on
[13]	**2021**	Overview of the academic research on the condition monitoring of rail transport systems	Bibliometric analysis based on content-based analysis, systematic literature review (SLR)	Web of Science, Scopus	316 papers from 1980–2020	Monitoring of railway transport systems as complex systems composed of various facilities and subsystems, discussing both rail tracks and rail vehicles
[14]	**2022**	Summary of monitoring, operation, and maintenance of offshore wind farms	Literature review	n/a	n/a	Offshore wind power engineering and biological and environment
[15]	**2019**	A comprehensive literature review on PdM with emphasis on system architectures, purposes, and approaches	Literature review	n/a	Papers from 2015–2019	Mainly ML-based and DL-based approaches
[16]	**2019**	Presentation of the literature review of ML methods applied to PdM with a particular focus on their main results, challenges, and opportunities	Systematic literature review (SLR)	IEEE Xplore, ScienceDirect	28 papers from 2009–2018	Current state-of-the-art for solutions of PdM techniques based on machine learning methods
[17]	**2021**	Review of the current literature concerning PdM and intelligent sensors in smart factories	Burst analysis, systematic review, co-occurrence analysis of keywords, and cluster analysis	Web of Science, Scopus	26 papers from 2010–2020	Intelligent sensors used for predictive maintenance in smart factories
[18]	**2022**	Review of methods and applied tools for intelligent PdM models in Industry 4.0	Literature review	n/a	n/a	Models associated with this type of maintenance: CBM, PHM, and RUL
[19]	**2022**	The analysis of existing ontology evolution methodologies and their use in the field of predictive maintenance (PdM)	Systematic literature review (SLR)	Web of Science, ACM Digital Library, IEEE Xplore	140 papers from 2017–2022	Time-sensitive domains and knowledge-based approach
[20]	**2022**	Review of VR, AR, and MR technologies and applications for smart building operation and maintenance	Literature review	Scopus, Web of Science, and Google Scholar	86 papers from 2018–2022	XR applications in the AECO industry
[21]	**2022**	Synthesizing the existing evidence on the application PdM with visual aids and identifying the key knowledge gaps in the investigated research area	Research Questions, a brief exploratory study	Web of Science, Scopus, IEEE Xplore	37 papers from 2017–2022	Implementation studies that utilized PdM to optimize utilities, power production, manufacturing, and energy consumption, and studies with human-centered data visualization methods
[22]	**2020**	Identification and analysis of frameworks, architectures, and tools in the area of predictive maintenance in Industry 4.0	Systematic literature review (SLR)	IEEE Xplore, Google Scholar, Springer, ACM Digital Library, ScienceDirect	38 papers from 2015–2020	Combination of ontologies, machine learning, and PdM
[23]	**2022**	Review of fault detection systems using the data collected from sensor devices/physical devices of various systems for PdM	Systematic literature review (SLR)	Scopus	93 papers from 2017–2021	Fault detection algorithms, anomaly detection
[24]	**2021**	Review on major expectations, requirements, and challenges for SMEs regarding the implementation of PdM	Systematic literature review (SLR)	IEEE Xplore, Springer	36 papers from 2010–2020	Smart manufacturing—PdM based on Industry 4.0 use in small- and medium-sized enterprises
[25]	**2022**	Overview of the current state of research concerning the PdM process from a data mining perspective	Systematic literature review (SLR)	Web of Science, Scopus, Google Scholar	132 papers from 2015–2021	Predictive maintenance, CBM, prognostic health management, data mining, machine learning and deep learning
[26]	**2018**	To investigate the role of maintenance for sustainable manufacturing, with a particular focus on the Industry 4.0 and the enabling technologies 4.0	Scoping literature review	Web of Science, Scopus	68 papers from 2003–2017	Industrial maintenance for sustainability in the Industry 4.0 context
[27]	**2022**	Review of maintenance employees’ competencies concerning Maintenance 4.0 characteristics and existing skills in Industry 4.0	Systematic literature review (SLR)	Google Scholar, Science Direct, and IEEE	52 papers from 2015–2020	19 competencies of Operator 4.0
[28]	**2022**	Performance of a bibliometric study to analyze and quantify the most important concepts, application areas, methods, and main trends of AI applied to real-time predictive maintenance	Bibliometric performance analysis	Web of Science	4065 papers from 2000–2021	Guidelines that may help researchers and practitioners to understand the key challenges and the most insightful scientific issues that characterize a successful application of AI to PdM4.0
[29]	**2018**	Review of machine health management for the smart factory	Literature review	n/a	97 papers from 1993–2017	Different types of machine health managements techniques in terms of data connectivity, communications, CPS and virtual factory, IoT, cloud computing, and big data management
[30]	**2021**	Review on PdM in relation to the exploration of machine learning and deep learning algorithms to improve the performance of failure classification and detection	Systematic literature review (SLR)	IEEE Xplore, ScienceDirect, Springer, ACM, Research Gate, AAAI, Proc. of Science	32 papers from 2010–2021	Artificial intelligence algorithms to predict failures in mission-critical environments for supercomputing and deep learning techniques
[31]	**2021**	Review of PdM in smart grid distribution networks	Systematic literature review (SLR)	Scopus, ScienceDirect, IEEE Xplore, and Web of Science	65 papers from 2012–2020	Fault types and consequences, prediction methods and techniques
[32]	**2021**	Review on PdM using vibration analysis	Bibliometric review	Scopus	2086 papers from 2006–2021	Artificial Intelligence, machine learning, deep learning, Industry 4.0, data-driven model
[33]	**2020**	Review on current trends in diagnostics and prognostics for PdM	Systematic literature review (SLR)	IEEE Xplore, ScienceDirect, Springer, Web of Science	158 papers from 2015–2019	Predictive maintenance, condition-based maintenance, prognostics, and health management
[34]	**2020**	Review of recent technologies available in PdM with Industry 4.0 for SME	Literature review	n/a	n/a	Recent advancements in the IIoT with a corporative point-of-view
[35]	**2021**	Study of the evolution of concepts such as e-maintenance (eM) and intelligent maintenance (IM), together with emergent concepts such as smart maintenance (SM) and maintenance 4.0	Comparative review based on SLR, bibliometric analysis, a multiple case study, and experts survey	Scopus	773 papers from 1985–2020	Four concepts were selected to be investigated: e-maintenance, intelligent maintenance, smart maintenance, and Maintenance 4.0
[36]	**2022**	A review of PdM focused on a defense domain context, with a particular focus on the operations and sustainment of fixed-wing defense aircraft	Systematic literature review (SLR)	Scopus	50 papers from 2000–2022	PdM with military applications
[37]	**2022**	Review of Industry 4.0 technologies used in maintenance management	Literature review	Web of Science, Scopus, and Google Scholar	54 papers from 2017–2022	Integration of the main functions and components of the maintenance management model and the Industry 4.0 features and technologies
[38]	**2020**	Analysis of maintenance tasks and maintenance management strategies development in Industry 4.0 context	Systematic literature review (SLR)	Scopus, IEEE Xplore, Google Scholar, Web of Science	65 papers from 2015–2019	The state-of-the-art Industry 4.0 technologies currently employed in the maintenance
[39]	**2022**	Study on the challenges of the PdM	Systematic literature review (SLR)	Google Scholar	91 papers from 2016–2021	Predictive models, engineering, prognostic and health management, remaining useful life and CBM
[40]	**2022**	Overview of studies on PdM using digital twins	Systematic literature review (SLR)	ScienceDirect, Scopus, ACM Digital Library, IEEE Xplore, Wiley, Taylor and Francis Online, Springer Link	42 papers from 2002–2021	Digital Twin in predictive maintenance, predictive maintenance system, cyber–physical system
[41]	**2020**	Review on smart remanufacturing and maintenance in the era of Industry 4.0	Literature review	Scopus, ScienceDirect, and ProQuest	2000–2020	Automated inspection, condition monitoring, and integrated optimization of production and maintenance planning
[42]	**2020**	Review on PdM in Industry 4.0 in the context of identifying and cataloging methods, standards, and applications	Systematic literature review (SLR)	Google Scholar, Association for Computing Machinery (ACM), IEEE, ScienceDirect, Scopus, Web of Science	47 papers from 2008–2018	Prediction or monitoring applied to Industry 4.0, smart factory, IoT as a model, method, or architecture

**Table 2 sensors-23-01409-t002:** A summary of the conducted filtering procedure.

Combination of Words for Inclusion Criteria	Research Results
Maintenance 4.0 AND Data-driven decision	36
Maintenance 4.0 AND System architecture	165
Maintenance 4.0 AND Operator 4.0	25
Maintenance 4.0 AND Virtual reality	182
Maintenance 4.0 AND Augmented reality	272
Maintenance 4.0 AND Cybersecurity	127
**Total (records selected for further analysis)**	**807**

## Data Availability

Not applicable.

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
