# Peer review of "Maintenance Performance in the Age of Industry 4.0: A Bibliometric Performance Analysis and a Systematic Literature Review"

_sensors, 2023, doi:10.3390/s23031409_

Round 1

Reviewer 1 Report (Previous Reviewer 2)

Author Response

I enclose the revised manuscript titled Maintenance Performance in the Age of Industry 4.0 - A Bibliometric Performance Analysis and a Systematic Literature Review, prepared by authors: Sylwia Werbińska-Wojciechowska and Klaudia Winiarska to be reviewed and published in Sensors Special Issue "Intelligent Control and Digital Twins for Industry 4.0". Manuscript ID: sensors-2162226.

We are very thankful to the Reviewer for his/her valuable comments and suggestions to our paper "Maintenance Performance in the Age of Industry 4.0 - A Bibliometric Performance Analysis and a Systematic Literature Review". We have revised the manuscript considering the valuable inputs from the Reviewer. The responses to the comments from the Reviewer are submitted in the submission system. All the revisions that have been made are clearly highlighted in the text (we have used the "Track Changes" function) and indicated in the responses for Reviewer.

The detailed responses for Reviewer comments are attached as a separate file.

Reviewer 2 Report (New Reviewer)

Thank you for the opportunity to review your manuscript. I appreciate your efforts in the work put into this extensive research Maintenance Performance in the Age of Industry 4.0 – A Bibliometric Performance Analysis and a Systematic Literature Review:

-        The information is easy to navigate, and the graphic structure of the paper allows readers to analyze the concepts approached, providing an interesting insight of the topic.

-        The paper is well written according to academic standards, using proper language and scientific style.

-        The literature review provides a good background of the topic.

-        The authors bring relevant and interesting arguments.

-        The findings and also the conclusions are in line with the results of the theoretical evidence. 

Although, to enhance the quality of the study, it would be wise to pay attention to several issues:

-        The presented study is relatively voluminous (64 pages). Despite the very broad use of sources of information, it should be considered to make the study more compact, especially in the descriptive (theoretical) areas and table formatting, which will certainly allow better focus on the main objectives and results.

-        Some figures could be resized to shorten the number of pages (for example: Figure 3, 8, 9, 11, 12, 20, 21 etc). Figure 19 could be eliminated, as it does not brig much value to the content.

-        Authors should include some recent works that use bibliometric review, such as:

Bibliometric Analysis of the Green Deal Policies in the Food Chain. Amfiteatru Econ. 2022, 24, 410–428. DOI:10.24818/EA/2022/60/410.

Mapping Knowledge Area Analysis in E-Learning Systems Based on Cloud Computing. Electronics 2023, 12, 62. https://doi.org/10.3390/electronics12010062.

Exploring the Research Regarding Frugal Innovation and Business Sustainability through Bibliometric Analysis. Sustainability. 2022, 14(3), 1326. https://doi.org/10.3390/su14031326.

-        The reference list is not formatted according to the MDPI guidelines. Authors should correct this aspect.

Good luck with your revision!

Author Response

I enclose the revised manuscript titled Maintenance Performance in the Age of Industry 4.0 - A Bibliometric Performance Analysis and a Systematic Literature Review, prepared by authors: Sylwia Werbińska-Wojciechowska and Klaudia Winiarska to be reviewed and published in Sensors Special Issue "Intelligent Control and Digital Twins for Industry 4.0". Manuscript ID: sensors-2162226.

We are very thankful to the Reviewer for his/her valuable comments and suggestions to our paper "Maintenance Performance in the Age of Industry 4.0 - A Bibliometric Performance Analysis and a Systematic Literature Review". We have revised the manuscript considering the valuable inputs from the Reviewer. The responses to the comments from the Reviewer are submitted in the submission system. All the revisions that have been made are clearly highlighted in the text (we have used the "Track Changes" function) and indicated in the responses for Reviewer.

The detailed responses for Reviewer comments are attached as a separate file.

Reviewer 3 Report (Previous Reviewer 3)

Dear author(s),

Thank you very much for the opportunity to review your manuscript.

This manuscript has potential, but the manuscript needs to improve substantially in terms of clarity both regarding results and methodological procedures to be considered for publication. There are several statements in these sections that need to be clearly explained. In addition, an English review is highly recommended as there are several unusual words/expressions used and sentences difficult to understand. Please see below some comments/suggestions for your evaluation if they can contribute to improve it.

Please consider improving the abstract regarding the “inclusion criteria” as it is not clear how these “research fields” were used as inclusion criteria. It should be clear how these “criteria” were evaluated.

It is not clear what was the contribution of the authors in Figure 2 and what was taken from references 41 and 42. The same applies to all other figures that the authors mention there is their “own contribution”. This needs to be clarified to avoid questions about authorship and to clearly inform the reader what was developed by the authors and what was developed by the cited reference.

When the authors mention the "search engines" used, they put them inside squares, which are not textual elements such as tables or figures.

The authors need to be more clear regarding the “search engine” related to WoS. Please state what are the numbers accompanying words. There is no explanation about this.

The authors state that 1113 records from the Web of Science database and 1933 from the Scopus database were considered in Phase I. There is no explanation of:

·         how many duplicate documents were found;

·         how these two bases were merged;

·         what was the final sample considered.

The search string contains subject area filters, but there is no explanation on ‘what’ and ‘why’.

Citation format needs to be revised throughout the text (e.g. line 439).

Please check capitalized words (e.g. Subsections in line 444).

A central question that require clear explanation is how the authors defined “six inclusion criteria”. These are “research fields” or “research topics”, and it is not clear how they are used as “inclusion criteria” and how the authors arrived at this “result” (as they claim that this resulted from the first phase of the study – see line 465).

Please consider the following: “Out of the initial 219 725 records, 218 918 were eliminated during the screening process.” – Please explain how the screening process of these 219,725 documents was conducted – i.e., was this done manually? Was there any help from any software? How many analysts participated in this stage of the study?

Please consider the following: “As a result of the filtering process taking into account the inclusion criteria, 218 918 were eliminated out of the initial 219 725 records.” – Please consider informing the resulting amount of documents, saving the reader from doing the math.

What is meant by “unique studies” (please to refer to line 662)? This raises the issue of originality, as every published article is presumably original. The authors need to be clear about the terms and expressions they use.

The three most published sources consist of conference proceedings (IFAC-PapersOnLine, Procedia Computer Science, and Procedia CIRP). Just these three sources account for 78.5% (168 out of 214) of the sample. This issue needs to be discussed by the authors as they state in the methodology section that limited the documents to “articles, books, and book chapters for higher quality of data.” This weakens the results.

The authors state that “The investigated 214 papers include 147 articles published in scientific journals, 34 book chapters and 33 conference papers published in international conference materials” – However, as previously indicated, at least 78.5% of the sample consists of conference proceedings. For instance, Figure 21 shows Procedia CIRP in first place – the authors can verify in their website that “Procedia CIRP is an open access product focusing entirely on publishing high quality proceedings from CIRP conferences” (please refer to https://www.sciencedirect.com/journal/procedia-cirp).

Author Response

I enclose the revised manuscript titled Maintenance Performance in the Age of Industry 4.0 - A Bibliometric Performance Analysis and a Systematic Literature Review, prepared by authors: Sylwia Werbińska-Wojciechowska and Klaudia Winiarska to be reviewed and published in Sensors Special Issue "Intelligent Control and Digital Twins for Industry 4.0". Manuscript ID: sensors-2162226.

We are very thankful to the Reviewer for his/her valuable comments and suggestions to our paper "Maintenance Performance in the Age of Industry 4.0 - A Bibliometric Performance Analysis and a Systematic Literature Review". We have revised the manuscript considering the valuable inputs from the Reviewer. The responses to the comments from the Reviewer are submitted in the submission system. All the revisions that have been made are clearly highlighted in the text (we have used the "Track Changes" function) and indicated in the responses for Reviewer.

The detailed responses for Reviewer comments are attached as a separate file.

Round 2

Reviewer 3 Report (Previous Reviewer 3)

Dear authors,

Thank you very much for the revised version of your manuscript.

I believe the points of concern have been addressed. The points that raised questions were improved and/or clarified. The changes made by the authors resulted in significant improvement in the manuscript.

This manuscript is a resubmission of an earlier submission. The following is a list of the peer review reports and author responses from that submission.

Round 1

Reviewer 1 Report

Detected 8% coincidences with this paper

https://www.mdpi.com/2076-3417/10/15/5172

Reviewer 3 Report

Dear author(s),

Thank you very much for the opportunity to review your manuscript. The authors' efforts in synthesizing the literature should be acknowledged, but the manuscript needs to improve substantially in terms of clarity and methodological rigor. Readability and clarity are highly problematic points. There are numerous confusing textual passages, conflicting information, statements lacking references or conceptual/statistical support, and methodological procedures lacking explanation. Please see below some comments/suggestions for your evaluation if they can contribute to improve it.

The abstract is missing results, i.e., which are the "five groups" defined, which are "the main research 40 problems and trends", etc.

The abstract needs clarification regarding the “inclusion criteria” as it is not clear how these “topics” were used as inclusion criteria (that usually include study characteristics, research area, etc.)

The first paragraph of the Introduction is missing references for the statements made.

The Introduction contains some loose sentences without any explanation / debate / argumentation, for example, "Known solutions have evolved from Maintenance 1.0 to Maintenance 4.0"

The entire text must be thoroughly proofread to correct errors such as “Additionally, the presented in Figure X…”

Please consider the following: “Based on the information in key scientific databases, more than 50 review papers on predictive maintenance/Maintenance 4.0 have been published in the last decade” - On what basis is such a statement made? Which databases were considered?

Regarding Table 1: what is the difference between “systematic literature review” and “systematic review”?

What is meant by “Objective research”?

The Theoretical Background contains several statements that require reference. In fact, all paragraphs have sentences that need references. This needs to be revised.

It is not clear what was the contribution of the authors in Figure 2 and what was taken from references 26 and 27. The same applies to all other figures that the authors mention there is their “own contribution”. This needs to be clarified to avoid questions about authorship and to clearly inform the reader what was developed by the authors and what was developed by the cited reference.

“OEE” needs to be explain to the reader in its first use. The same applies to other abbreviations.

Please consider the following: “In addition, the search was limited to crucial citation topics related to production, logistics or maintenance (non-topically related publications were excluded)” – There is a need to clarify what the authors mean by “crucial citation topics”, “non-topically related publications”, etc. In addition, there is a need to explain how this analytical procedure was done, i.e., how the 4944 documents were analyzed.

The authors need to be more clear regarding lines 349-353. Please explain what are these numbers and what is “citation topics meso”.

The search string contains subject area filters, but this is not explained in the text.

Citation format in the line 378 needs to be fixed.

“)” is missing in the line 385

The authors state that “the investigation was completed in September 2022…”; then, they state that “the literature search was conducted between 2 October 2022 and 10 October 2022” (sic).

“ is missing in the line 393

Please consider the following: “These criteria have been established based on the authors' experience and reflect the most relevant aspects of the analyzed research area.” - Although the authors' experience is valuable, the method of systematic literature review seeks precisely to rely on objective, demonstrable and verifiable criteria. In this sense, this aspect weakens the methodology.

Please consider the following: “Out of the initial 219 725 records, 218 918 were eliminated during the screening process.” – Please explain how the screening process of these 219,725 documents was conducted.

What is meant by “unique approaches” (please to refer to line 407)

What is meant by “wrong publication type”?

Please consider the following: “This choice was dictated by the fact that these are the most important online databases with full access availability for authors” – Based on what criteria this sentence is made? In what aspects Springer or Elsevier are “most important” than other databases such as Emerald, Scopus or Google Scholar?

The authors state that “As a result, there were defined 486 papers, which were later fully read to identify the most relevant papers.” – How was relevance determined? How many researchers participated in this process? How were disagreements resolved?

After section 3.1, section 2.2 appears – please verify.

The authors predict (sic) that 92 papers will be published on Scopus and 40 papers will be published on Web of Science – i) there is no basis for this “prediction”; ii) this is not consistent with the data presented.

The authors state that they focused on “journal articles, books, and book chapters”. However, the three most published sources consist of conference proceedings (IFAC-PapersOnLine, Procedia Computer Science, and Procedia CIRP). Just these three sources account for 78.5% (168 out of 214) of the sample.

There is no discussion on Figures 12 to Figure 17.

The authors state that publications their results show a “clearly growing [publication] trend from 2017.” – However, Figure 19 shows a decrease from 2017 to 2018, from 2019 to 2020, and from 2021 to 2022. This is very confusing.

The authors state that “The investigated 214 papers include 147 articles published in scientific journals, 34 book chapters and 33 conference papers published in international conference materials” – However, as previously indicated, at least 78.5% of the sample consists of conference proceedings.

Figure 20 showed very different data from what was described earlier in the text. This is very confusing.

Section 6 is named “Conclusions with discussion”, which is quite unusual. Please consider differentiating “discussions” from “conclusions” as they have very different roles in a paper.